# MAP IT to Visualize Representations

**Robert Jenssen**

UiT The Arctic University of Norway & U Copenhagen & Norwegian Computing Center

## Abstract

*MAP IT* visualizes representations by taking a fundamentally different approach to dimensionality reduction. MAP IT aligns distributions over discrete marginal probabilities in the input space versus the target space, thus capturing information in wider local regions, as opposed to current methods which align based on pairwise probabilities between states only. The MAP IT theory reveals that alignment based on a projective divergence avoids normalization of weights (to obtain true probabilities) entirely, and further reveals a dual viewpoint via continuous densities and kernel smoothing. MAP IT is shown to produce visualizations which capture class structure better than the current state of the art.

## 1 Introduction

Representation learning is key to any machine learning system. For instance for learning to visualize input representations, or for visualizing learned representations obtained e.g. via deep neural networks trained in a supervised or unsupervised/self-supervised manner in order to gain insight[1].

Methods such as t-SNE (Böhm et al., 2023; Tucker et al., 2023; Huang et al., 2022; van der Maaten & Hinton, 2008; van der Maaten, 2014), the closely related LargeVis (Tang et al., 2016), as well as the recently proposed UMAP (McInnes et al., 2020), TriMAP (Amid & Warmuth, 2019) and PacMap (Wang et al., 2021), are tremendously important contributions, spurring numerous theoretical works, e.g. (Draganov et al., 2023; Damrich & Hamprecht, 2021; Kobak & Linderman, 2021).

However, these dominant dimensionality reduction (DR) methods (further discussed in Appendix A) take vastly different theoretical approaches, but despite this diversity, none of the theories provide a framework which explain when normalization of weights is needed and when not. Understanding the role of normalization in DR has been an important quest (Draganov et al., 2023). Moreover, all existing methods are based on pairwise comparisons of weights in the input space and the target space without considering wider information from local regions.

MAP IT represents a major shift. The MAP IT framework is based on information theory (IT) and statistical divergences, showing that normalization is not needed when the divergence used is *projective*. Based on a Cauchy-Schwarz projective divergence, MAP IT aligns distributions of marginal probabilities which capture information about wider local regions in the data, in essence aggregating information from pairwise weights in a new manner. A dual viewpoint to MAP IT is derived via continuous densities enabled by kernel smoothing, revealing the role of neighborhoods for the gradient-based MAP IT. See Appendix B for all proofs with additional comments.

MAP IT creates overall markedly different embeddings. As an example, Figure 1 shows MAP IT embeddings of a subset of MNIST (d) compared to t-SNE (a), UMAP (b) and PacMAP (c). MAP IT separates four of the classes of digits much clearer than any of the other methods. For the challenging digits 4, 9 and 7 (rectangles, zoomed), MAP IT creates less overlap between 4s and 9s, and much better separation of 7s (f) compared to e.g. PacMap (e).

The main contributions of this paper are i) Providing a new theory for visualization by dimensionality reduction; ii) Showing that normalization of weights to obtain true probabilities is not needed in MAP IT by the use of a projective divergence; iii) Revealing a dual viewpoint by aligning over distributions over discrete marginal probabilities or equivalently via continuous densities enabled by kernel smoothing; iv) Shedding light on the role of local neighborhoods for the MAP IT update rule; v) Generating markedly different visualizations compared to the current state of the art.

---

[1]To prepare the representations for a downstream task can also be of interest but not considered here.

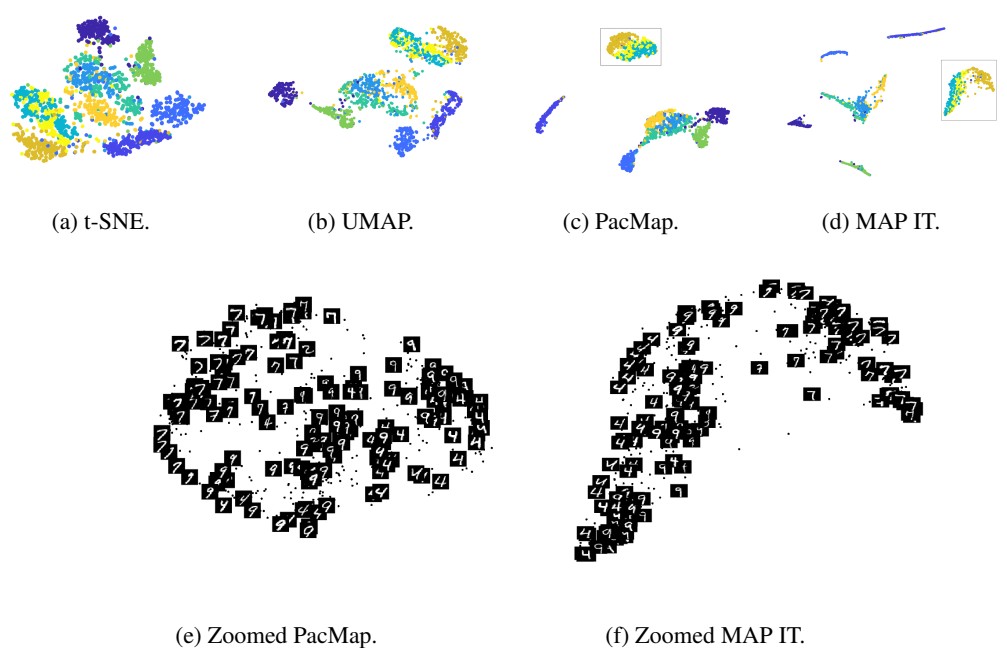

(a) t-SNE.  (b) UMAP.  (c) PacMap.  (d) MAP IT.

(e) Zoomed PacMap.  (f) Zoomed MAP IT.

Figure 1: Visualizing a subset of MNIST. MAP IT separates digits much more distinctly compared to the alternatives and creates less overlap on difficult classes as shown when zooming in (rectangles).

## 2   t-SNE FRAMEWORK

t-SNE minimizes the Kullback-Leibler (KL) divergence between a joint probability distribution $P$ over states given by $\boldsymbol{x}_1, \ldots, \boldsymbol{x}_n \in \mathbb{R}_D$ in the high-dimensional input space and a joint probability distribution $Q$ over states (mapped data points) $\boldsymbol{z}_1, \ldots, \boldsymbol{z}_n \in \mathbb{R}_d$ in the low-dimensional target space

$$\underset{\boldsymbol{z}_1,\ldots,\boldsymbol{z}_n \in \mathbb{R}_d}{\arg\min} KL(P||Q) = \underset{\boldsymbol{z}_1,\ldots,\boldsymbol{z}_n \in \mathbb{R}_d}{\arg\min} \sum_{i,j} p_{ij} \log \frac{p_{ij}}{q_{ij}} = \underset{\boldsymbol{z}_1,\ldots,\boldsymbol{z}_n \in \mathbb{R}_d}{\arg\min} -\sum_{i,j} p_{ij} \log q_{ij}, \quad (1)$$

where $q_{ij} = \left[1 + ||\boldsymbol{z}_i - \boldsymbol{z}_j||^2\right]^{-1} / Z_q$ are joint probabilities in the target space defined by the t-distribution (Cauchy distribution) and where $Z_q = \sum_{n,m} \left[1 + ||\boldsymbol{z}_n - \boldsymbol{z}_m||^2\right]^{-1}$ is an explicit normalization factor. Note that for this particular choice, one may write $q_{ij} = \tilde{q}_{ij}/Z_q$ where $\tilde{q}_{ij} = \left[1 + ||\boldsymbol{z}_i - \boldsymbol{z}_j||^2\right]^{-1}$ is an unnormalized quantity. To better prepare for the theory outlined in Section 3, the model $p_{ij} = \exp\left(-\kappa_i ||\boldsymbol{x}_i - \boldsymbol{x}_j||^2\right) / Z_p$ will be assumed, with the normalization factor $Z_p = \sum_{n,m} \exp(-\kappa_i ||\boldsymbol{x}_n - \boldsymbol{x}_m||^2)$ (further discussed in Appendix C).

**Proposition 1.** *[Minimizing $KL(P||Q)$ and the role of normalization]. Let $q_{ij} = \tilde{q}_{ij}/Z_q$ for some choice of $\tilde{q}_{ij}$ where $Z_q = \sum\limits_{n,m} \tilde{q}_{nm}$. Then*

$$\underset{\boldsymbol{z}_1,\ldots,\boldsymbol{z}_n \in \mathbb{R}_d}{\arg\min} KL(P||Q) = \underset{\boldsymbol{z}_1,\ldots,\boldsymbol{z}_n \in \mathbb{R}_d}{\arg\min} -\sum_{i,j} p_{ij} \log \tilde{q}_{ij} + \log \sum_{n,m} \tilde{q}_{nm}. \quad (2)$$

**Comment to Proposition 1.** The t-SNE cost function may by this result be expressed both over probabilities $p_{ij}$ and over quantities $\tilde{q}_{ij}$ in low dimensional space which are not probabilities as they are not normalized[2]. This also carries over to the expression for the gradient of $KL(P||Q)$:

---

[2]Some interesting properties of the t-SNE cost function related to Laplacian eigenmaps may be highlighted when inserting the explicit form for $\tilde{q}_{ij}$. This is explored in Appendix A.

**Proposition 2.** *[Gradient of $KL(P||Q)$]*

$$\frac{\partial}{\partial \boldsymbol{z}_i} KL(P||Q) = -4 \sum_j \left( p_{ij} - q_{ij} \right) \tilde{q}_{ij} (\boldsymbol{z}_j - \boldsymbol{z}_i). \tag{3}$$

**The role of normalization.** In t-SNE, gradient descent can only be performed after computing $q_{ij}$ over the entire data set for each iteration or epoch, and can be interpreted as an $n$-body system of attractive forces and repulsive forces. Speeding up these computations is the reason for the development of the Barnes-Hut tree algorithm (van der Maaten, 2014) or the more recent interpolation-based t-SNE (Linderman et al., 2019). The requirement for normalization also hampers the utility of parametric t-SNE (van der Maaten, 2009) as discussed in (Sainburg et al., 2021). Recently, new algorithms and insight to normalization have been revealed (Artemenkov & Panov, 2020; Damrich et al., 2023) from the viewpoint of noise contrastive learning (Gutmann & Hyvärinen, 2012), also in the context of the interplay between attractive and repulsive forces (Böhm et al., 2022).

**Brief perspectives on t-SNE, UMAP, PacMap and variants.** A number of new dimensionality reduction methods have been proposed, most being more computationally efficient than t-SNE, usually ending up as a system of attractive and repulsive forces. UMAP (McInnes et al., 2020) exploits cross-entropy between simplical sets and invokes a sampling strategy on repulsive forces which has resemblance to the approach proposed in LargeVis (Tang et al., 2016) (a method inspired by t-SNE). TriMAP (Amid & Warmuth, 2019) and PacMap (Wang et al., 2021) are heuristic methods where the former is based on triplet loss and the latter empirically sets up a cost function based on pairwise distances between near pairs, mid-near pairs, and non-neighbor pairs. These methods are considered to be leading approaches, elaborated in more detail in the Appendix A[3].

## 3 MAP IT FRAMEWORK

### 3.1 MAP IT BY THE CAUCHY SCHWARZ DIVERGENCE

SNE is elegantly formulated in terms of the KL divergence. However, the KL divergence requires a normalization factor since the KL divergence is not *projective*.

**Definition 1** (Projective Divergence). *Let $p_{ij} = \tilde{p}_{ij}/Z_p$ and $q_{ij} = \tilde{q}_{ij}/Z_q$ where $Z_p = \sum_{n,m} \tilde{p}_{nm}$ and $Z_q = \sum_{n,m} \tilde{q}_{nm}$ are normalizing constants. This means that $P = \tilde{P}/Z_p$ and $Q = \tilde{Q}/Z_q$. A projective divergence is such that*

$$D(P||Q) = D(\tilde{P}||\tilde{Q}) \tag{4}$$

*where $D$ is the divergence measure.*

For optimization purposes, the use of a projective divergence[4] function may be appealing since the direct dependence on normalization constants is removed. A divergence measure which has received widespread attention in recent years is the Cauchy-Schwarz divergence (Yu et al., 2023; 2022; Principe, 2010; Jenssen et al., 2006).

Without loss of generality, the CS divergence is here defined in terms of marginal probabilities $P(\boldsymbol{x}_j) = p_j$ in the input space and $P(\boldsymbol{z}_j) = q_j$ in the target space, respectively.

**Definition 2** (Cauchy-Schwarz (CS) Divergence). *Denote $P_m = \{p_j\}_{j=1}^N$ and $Q_m = \{q_j\}_{j=1}^N$. Then*

$$CS(P_m||Q_m) = -\log \frac{\sum_j p_j q_j}{\left(\sum_j p_j^2\right)^{\frac{1}{2}} \left(\sum_j q_j^2\right)^{\frac{1}{2}}}. \tag{5}$$

---

[3]A plethora of dimensionality reduction variants exist, see e.g. the overview (Wang et al., 2021) or visit the excellent repository `https://jlmelville.github.io/smallvis/`. t-SNE and the other methods mentioned here all use gradient descent but build on many aspects from spectral theory, e.g. (Roweis & Saul, 2000), (Tenenbaum et al., 2000), (Belkin & Niyogi, 2003), (Donoho & Grimes, 2003), (Jenssen, 2010).

[4]Could equivalently be defined with marginal probabilities $q_i = \tilde{q}_i/Z_q$, $Z_q = \sum_i \tilde{q}_i$, or continuous probability functions $q(\boldsymbol{z}) = \tilde{q}(\boldsymbol{z})/Z_q$, $Z_q = \int \tilde{q}(\boldsymbol{z}) d\boldsymbol{z}$ (likewise for $p(\boldsymbol{x})/p_i$).

**Proposition 3.** *[CS divergence is projective].* *Let* $p_j = \tilde{p}_j/Z_p$, $Z_p = \sum_{j'} \tilde{p}_{j'}$, *and* $q_j = \tilde{q}_j/Z_q$, $Z_q = \sum_{j'} \tilde{q}_{j'}$. *Then*

$$CS(P_m||Q_m) = CS(\tilde{P}_m||\tilde{Q}_m). \tag{6}$$

The CS divergence will provide the backbone for the MAP IT framework. MAP IT will be derived both from the viewpoint of marginal probabilities and from the viewpoint of a continuous risk function. The latter provides a coupling to kernel methods which provides a dual framework. MAP IT's cost function lends itself nicely to optimization with gradient descent, explained in the next sections.

## 3.2 MAP IT VIA MARGINAL PROBABILITIES

By the law of total probability $p_j = \sum_k p_{jk}$ and $q_j = \sum_k q_{jk}$. Since $p_{jk}$ and $q_{jk}$ are assumed normalized (Eq. (1)), $\sum_j p_j = 1$ and $\sum_j q_j = 1$. Note that by this argument one may express $q_j = \tilde{q}_j/Z_q$, where $\tilde{q}_j = \sum_k \tilde{q}_{jk}$, $Z_q = \sum_{n,m} \tilde{q}_{nm}$, and likewise for $p_j$.

**Definition 3** (MAP IT cost function). *Let* $\tilde{p}_j = \sum_k \tilde{p}_{jk}$ *and* $\tilde{q}_j = \sum_k \tilde{q}_{jk}$. *The proposed MAP IT cost function and optimization problem are*

$$\arg\min_{\boldsymbol{z}_1,\ldots,\boldsymbol{z}_n \in \mathbb{R}_d} CS(P_m||Q_m) = \arg\min_{\boldsymbol{z}_1,\ldots,\boldsymbol{z}_n \in \mathbb{R}_d} CS(\tilde{P}_m||\tilde{Q}_m). \tag{7}$$

**Comment to Definition 3.** A major deviation from the SNE framework is that MAP IT aims to align distributions over marginal probabilities for states corresponding to data points in the two spaces, i.e. $p_j$ and $q_j$. These probabilities can be interpreted as degrees associated with the nodes of the underlying similarity graphs in the two spaces, reflecting properties of regions of the data, as opposed to considering only pairwise local connections. This is associated with the underlying mapping which may be expressed as $\boldsymbol{z}_j = m(\boldsymbol{x}_j) + \varepsilon_j$ where $m(\cdot)$ is the mapping function and $\varepsilon_j$ is some noise random variable. Minimizing $CS(P_m||Q_m)$ yields:

**Proposition 4.** *[Minimizing $CS(P_m||Q_m)$ with respect to $\boldsymbol{z}_1,\ldots,\boldsymbol{z}_n \in \mathbb{R}_d$]*

$$\arg\min_{\boldsymbol{z}_1,\ldots,\boldsymbol{z}_n \in \mathbb{R}_d} CS(\tilde{P}_m||\tilde{Q}_m) = \arg\min_{\boldsymbol{z}_1,\ldots,\boldsymbol{z}_n \in \mathbb{R}_d} -\log \sum_j \tilde{p}_j \tilde{q}_j + \frac{1}{2}\log \sum_j \tilde{q}_j^2. \tag{8}$$

As a main result of this paper, the gradient of MAP IT, needed in order to perform minimization of $CS(P_m||Q_m)$ over $\boldsymbol{z}_1,\ldots,\boldsymbol{z}_n \in \mathbb{R}_d$, is derived and stated in the following proposition:

**Proposition 5.** *[Gradient of $CS(P_m||Q_m)$]*

$$\frac{\partial}{\partial \boldsymbol{z}_i} CS(P_m||Q_m) = -4 \sum_j \left[ \frac{\tilde{p}_j}{\sum_{j'} \tilde{p}_{j'}\tilde{q}_{j'}} - \frac{\tilde{q}_j}{\sum_{j'} \tilde{q}_{j'}^2} \right] \tilde{q}_{ij}^2 (\boldsymbol{z}_j - \boldsymbol{z}_i). \tag{9}$$

**Comment to Proposition 4.** The MAP IT update rule $\boldsymbol{z}_i = \boldsymbol{z}_i - \eta \frac{\partial}{\partial \boldsymbol{z}_i} CS(P_m||Q_m)$ yields

$$\boldsymbol{z}_i = \boldsymbol{z}_i + \eta \sum_j \left[ \frac{\tilde{p}_j}{\sum_{j'} \tilde{p}_{j'}\tilde{q}_{j'}} - \frac{\tilde{q}_j}{\sum_{j'} \tilde{q}_{j'}^2} \right] \tilde{q}_{ij}^2 (\boldsymbol{z}_j - \boldsymbol{z}_i) \tag{10}$$

where the factor *four* has been fused into $\eta$. MAP IT thus has easily interpretable gradients, forming a $n$-body problem where each $\boldsymbol{z}_i$ experiences an attractive force $\tilde{p}_j$ and a repulsive force $\tilde{q}_j$ from each other data point $\boldsymbol{z}_j$. The forces are adaptively weighted by entropy in the sense that $-\log \sum_{j'} \tilde{q}_{j'}^2$ is a Renyi entropy of $Q_m$ and $-\log \sum_{j'} \tilde{p}_{j'}\tilde{q}_{j'}$ is a cross entropy. All quantities are unnormalized.

## 3.3 MAP IT VIA CONTINUOUS DENSITY FUNCTIONS

This section provides an alternative derivation for MAP IT via kernel smoothing. This reveals the role of local neighborhoods via derivation, affecting the computation of gradients. The following definition is provided, referred here to as the *map* CS divergence:

**Definition 4** (*Map* CS divergence). *Let $p(\boldsymbol{x})$ be the probability density function over a domain $\mathcal{X}$ and define $q(\boldsymbol{z})$ to be the probability density function for the stochastic map $\boldsymbol{z} = m(\boldsymbol{x}) + \varepsilon$ over $\mathcal{Z}$. Let $f(\boldsymbol{x}, \boldsymbol{z})$ be the joint distribution over $\mathcal{X} \times \mathcal{Z}$. The* map *CS divergence is defined as*

$$CS(p(\boldsymbol{x})||q(\boldsymbol{z})) = -\log \frac{\int \int p(\boldsymbol{x})q(\boldsymbol{z})f(\boldsymbol{x}, \boldsymbol{z})d\boldsymbol{x}d\boldsymbol{z}}{\left(\int \int p^2(\boldsymbol{x})f(\boldsymbol{x}, \boldsymbol{z})d\boldsymbol{x}d\boldsymbol{z}\right)^{\frac{1}{2}} \left(\int \int q^2(\boldsymbol{z})f(\boldsymbol{x}, \boldsymbol{z})d\boldsymbol{x}d\boldsymbol{z}\right)^{\frac{1}{2}}}. \quad (11)$$

**Comment to Definition 4.** Consider two functions $g(\boldsymbol{x})$ and $h(\boldsymbol{z})$ where $\boldsymbol{z}$ is some mapping of $\boldsymbol{x}$. Within risk minimization, $R = E_{\mathcal{X} \times \mathcal{Z}} L[g(\boldsymbol{x}), h(\boldsymbol{z})] = \int \int L[g(\boldsymbol{x}), h(\boldsymbol{z})] f(\boldsymbol{x}, \boldsymbol{z})d\boldsymbol{x}d\boldsymbol{z}$ for some loss $L(\cdot)$. Empirical risk minimization aims to minimize $\hat{R} = \sum_i L[g(\boldsymbol{x}_i), h(\boldsymbol{z}_i)]$ with respect to some variables or parameters. From this, the map CS divergence may be understood as a normalized risk function over $g(\boldsymbol{x}) = p(\boldsymbol{x})$ and $h(\boldsymbol{z}) = q(\boldsymbol{z})$ with product loss.

**Proposition 6.** *[The* map *CS divergence is projective]. Let $\boldsymbol{z} = m(\boldsymbol{x}) + \varepsilon$ and assume $p(\boldsymbol{x}) = \tilde{p}(\boldsymbol{x})/Z_p$ and $q(\boldsymbol{z}) = \tilde{q}(\boldsymbol{z})/Z_q$ where $\tilde{p}(\boldsymbol{x})$ and $\tilde{q}(\boldsymbol{z})$ are unnormalized with $Z_p = \int \tilde{p}(\boldsymbol{x})d\boldsymbol{x}$ and $Z_q = \int \tilde{q}(\boldsymbol{z})d\boldsymbol{z}$ as the respective normalization constants. Then*

$$CS(p(\boldsymbol{x})||q(\boldsymbol{z})) = CS(\tilde{p}(\boldsymbol{x})||\tilde{q}(\boldsymbol{z})). \quad (12)$$

Assume in the following unnormalized functions $\tilde{p}(\boldsymbol{x})$ and $\tilde{q}(\boldsymbol{z})$.

**Proposition 7.** *[Empirical* map *CS divergence]. Let a sample $\boldsymbol{x}_1, \ldots, \boldsymbol{x}_n \in \mathbb{R}_D$ be given and assume the mapping $\boldsymbol{z} = m(\boldsymbol{x}) + \varepsilon$. Then, an empirical* map *CS divergence is given by*

$$\widehat{CS}(p(\boldsymbol{x})||q(\boldsymbol{z})) = -\log \frac{\sum_j \tilde{p}(\boldsymbol{x}_j)\tilde{q}(\boldsymbol{z}_j)}{\left(\sum_j \tilde{p}^2(\boldsymbol{x}_j)\right)^{\frac{1}{2}} \left(\sum_j \tilde{q}^2(\boldsymbol{z}_j)\right)^{\frac{1}{2}}}. \quad (13)$$

**Proposition 8.** *[Minimizing $\widehat{CS}(p(\boldsymbol{x})||q(\boldsymbol{z}))$ with respect to $\boldsymbol{z}_1, \ldots, \boldsymbol{z}_n \in \mathbb{R}_d$]*

$$\underset{\boldsymbol{z}_1, \ldots, \boldsymbol{z}_n \in \mathbb{R}_d}{\arg\min} \widehat{CS}(p(\boldsymbol{x})||q(\boldsymbol{z})) = \underset{\boldsymbol{z}_1, \ldots, \boldsymbol{z}_n \in \mathbb{R}_d}{\arg\min} -\log \sum_j \tilde{p}(\boldsymbol{x}_j)\tilde{q}(\boldsymbol{z}_j) + \frac{1}{2}\log \sum_j \tilde{q}^2(\boldsymbol{z}_j). \quad (14)$$

**Proposition 9.** *[Gradient of $\widehat{CS}(p(\boldsymbol{x})||q(\boldsymbol{z}))$]*

$$\frac{\partial}{\partial \boldsymbol{z}_i}\widehat{CS}(p(\boldsymbol{x})||q(\boldsymbol{z})) = -\sum_j \left[\frac{\tilde{p}(\boldsymbol{x}_j)}{\sum_{j'} \tilde{p}(\boldsymbol{x}_{j'})\tilde{q}(\boldsymbol{z}_{j'})} - \frac{\tilde{q}(\boldsymbol{z}_j)}{\sum_{j'} \tilde{q}^2(\boldsymbol{z}_{j'})}\right] \frac{\partial}{\partial \boldsymbol{z}_i}\tilde{q}(\boldsymbol{z}_j). \quad (15)$$

**Comment to Proposition 9.** Eq. (15) is a more general result compared to Eq. (9). For a particular choice or estimator of $\tilde{p}(\boldsymbol{x}_j)$ and $\tilde{q}(\boldsymbol{z}_j)$, the basic MAP IT update rule $\boldsymbol{z}_i = \boldsymbol{z}_i - \eta\frac{\partial}{\partial \boldsymbol{z}_i}CS(P_m||Q_m)$ will equal $\boldsymbol{z}_i = \boldsymbol{z}_i - \eta\frac{\partial}{\partial \boldsymbol{z}_i}\widehat{CS}(p(\boldsymbol{x})||q(\boldsymbol{z}))$ as is shown next. But this is not true in general, and the latter result may actually be more flexible.

Consider in the following the choice $\hat{\tilde{q}}(\boldsymbol{z}_j) = \sum_k \kappa_z(\boldsymbol{z}_j - \boldsymbol{z}_k)$ and similarly for $\hat{\tilde{p}}(\boldsymbol{x}_j)$. In this viewpoint, $\kappa(\cdot)$ is assumed to be a shift-invariant *kernel function* and $\sum_k \kappa_z(\boldsymbol{z}_j - \boldsymbol{z}_k)$ is the well-known kernel smoothing procedure, which is the backbone of reproducing kernel Hilbert space methods (e.g. (Schrab et al., 2023)) and of utmost importance in machine learning. Note that both the t-distribution (Cauchy distribution) and the Gaussian are valid kernel functions.

**Proposition 10.** *[Gradient of $\widehat{CS}(p(\boldsymbol{x})||q(\boldsymbol{z}))$ with kernel smoothing]. Let $\hat{\tilde{q}}(\boldsymbol{z}_j) = \sum_k \kappa_z(\boldsymbol{z}_j - \boldsymbol{z}_k)$ and $\hat{\tilde{p}}(\boldsymbol{x}_j) = \sum_k \kappa_p(\boldsymbol{x}_j - \boldsymbol{x}_k)$ for shift-invariant kernel functions $\kappa_z(\cdot)$ and $\kappa_p(\cdot)$. Then*

$$\frac{\partial}{\partial \boldsymbol{z}_i}\widehat{CS}(p(\boldsymbol{x})||q(\boldsymbol{z})) = -4\sum_j \left[\frac{\tilde{p}_j}{\sum_{j'} \tilde{p}_{j'}\tilde{q}_{j'}} - \frac{\tilde{q}_j}{\sum_{j'} \tilde{q}_{j'}^2}\right] \kappa_z^2(\boldsymbol{z}_j - \boldsymbol{z}_i)(\boldsymbol{z}_j - \boldsymbol{z}_i). \quad (16)$$

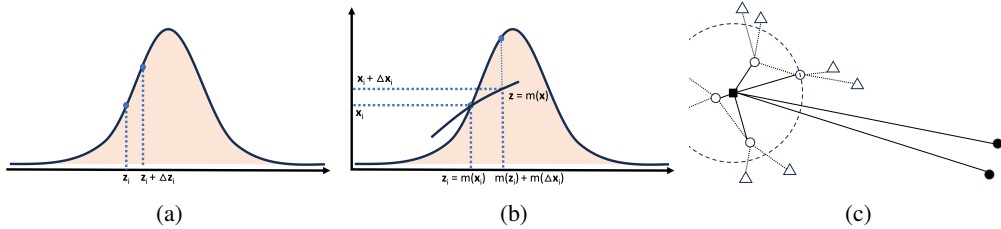

(a)  (b)  (c)

Figure 2: Illustrating the role of local neighborhoods in MAP IT.

**Comment to Proposition 10.** With the notation $\tilde{q}_{ji} \overset{\text{def}}{=} \kappa_z(z_j - z_i)$ associated with $\hat{\tilde{q}}(z_j) = \sum_k \kappa_z(z_j - z_k)$, this result shows that in this particular situation the MAP IT rule $z_i = z_i - \eta \frac{\partial}{\partial z_i} CS(P_m||Q_m)$ equals $z_i = z_i - \eta \frac{\partial}{\partial z_i} \widehat{CS}(p(x)||q(z))$.

MAP IT via marginal probabilities and MAP IT via continuous densities are thus *dual* viewpoints enabled by the map CS divergence and the kernel smoothing approach.

Having derived MAP IT, it is important to make clear that $t$-SNE based on the Cauchy-Schwarz divergence, instead of the Kullback-Leibler, appears as a special case of the theory.

**Proposition 11.** *[Cauchy-Schwarz (CS) t-SNE is a special case of MAP IT]. Let $p_{jk'}$ be the probability for the joint event $x_j \cap x_{k'}$. Let $q_{jk'}$ be the probability for the joint event $z_j \cap z_k$. If $(x_j \cap x_{k'}) \cap (z_j \cap z_k) \in \emptyset$, then*

$$CS(P_m||Q_m) = CS(P||Q). \tag{17}$$

### 3.4 MAP IT via Local Neighborhoods

One subtle but very important aspect which affects the MAP IT theory from the previous sections concerns the role of local neighborhoods. This is best revealed from the starting point of MAP IT via continuous densities as follows. The aim of MAP IT is to discover $z_1, \ldots, z_n \in \mathbb{R}_d$ from $x_1, \ldots, x_n \in \mathbb{R}_D$. This entails derivatives $\frac{\partial}{\partial z_i} \tilde{p}(\cdot)\tilde{q}(\cdot)$ and $\frac{\partial}{\partial z_i} \tilde{q}^2(\cdot)$ which are quantities defined over neighborhoods. Consider

$$\frac{\partial}{\partial z_i} \tilde{p}(\cdot)\tilde{q}(\cdot) \approx \frac{\tilde{p}(z_i + \Delta z_i)\tilde{q}(z_i + \Delta z_i) - \tilde{p}(z_i)\tilde{q}(z_i)}{\Delta z_i} \tag{18}$$

where $\Delta z_i$ defines a neighborhood in the target space around $z_i$. A sketch illustrating this situation is shown in Fig. (2) (a). However, an assumption is that an unknown transformation exists, $z = m(x) + \varepsilon$. If the transformation was known, there would be no need for MAP IT and neighborhoods $\Delta z_i$ around $z_i$ would correspond to neighborhoods $\Delta x_i$ around $x_i = m^{-1}(z_i)$. However, even if the transformation is not known, points in the two spaces come in pairs $z_i = m(x_i) + \varepsilon$, and over the course of the optimization neighborhood topologies in the two spaces should start to correspond. During the optimization, since the effect of the underlying mapping function only gradually will be learned, neighborhood topologies in the two spaces will not initially correspond. For differentiation with respect to $z_i$ one must therefore rely on $\Delta z_i = m(\Delta x_i)$. This is illustrated in Fig. (2) (b).

The practical consequence of the above analysis is that all marginal probabilities, part of the MAP IT learning rule, will be computed with respect to a neighborhood $m(\Delta x_i)$ and will be denoted $\tilde{p}_{j_{N_{x_i}}}$ and $\tilde{q}_{j_{N_{x_i}}}$, respectively, where $N_{x_i}$ refers to the neighborhood around $x_i$, yielding

$$z_i = z_i + \eta \sum_j \left[ \frac{\tilde{p}_{j_{N_{x_i}}}}{\sum_{j'} \tilde{p}_{j'} \tilde{q}_{j'}} - \frac{\tilde{q}_{j_{N_{x_i}}}}{\sum_{j'} \tilde{q}_{j'}^2} \right] \tilde{q}_{ij}^2 (z_j - z_i). \tag{19}$$

One may envision several possibilities for defining $N_{x_i}$. Here, it is argued that if $x_j$ is one of the nearest neighbors of $x_i$ then the nearest neighbors of $x_j$ are also i the neighborhood of $x_i$. The $\tilde{p}_{j_{N_{x_i}}}$ and $\tilde{q}_{j_{N_{x_i}}}$ are thus computed over the nearest neighbors of $x_j$. Of course, this line of thought

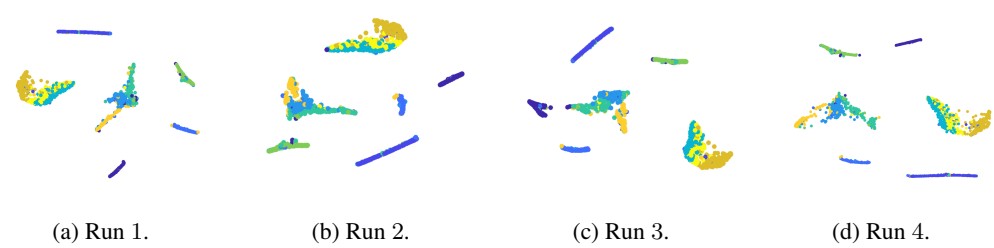

(a) Run 1.      (b) Run 2.      (c) Run 3.      (d) Run 4.

Figure 3: MAP IT for a subset of MNIST for different initial values (runs) for $k = 10$.

could have been continued to some degree (neighbors of neighbors), but is not explored further in this paper. Fig. (2) (c) shows as a filled black square a point $x_i$ ($z_i$) and how the nearest neighbors of $x_i$ as well as their neighbors constitute $N_{x_i}$. This resembles to some degree the use of near pairs and mid-near pairs in PacMap. For points that are not neightbors of $x_i$ one could compute $\tilde{p}_{j_{N_{x_i}}}$ and $\tilde{q}_{j_{N_{x_i}}}$ in a similar manner. However, since all $p_{ij}$ for $x_j$ not a neighbor of $x_i$ will be relatively small and constant with respect to points in the vicinity of $x_i$, only $p_{ij}$ ($q_{ij}$) is used in that case, as illustrated in Fig. (2) (c) (solid black circles). This resembles to some degree the use of non-near pairs in PacMac. Please see Appendix C for additional perspectives. Finding neighbors can be done in an exact manner, or by vintage trees (Yianilos, 1993), or by approximate search (Dong et al., 2011) as in LargeVis and UMAP.

Importantly, the MAP IT theory deviates fundamentally from the theory for t-SNE, UMAP, and all the alternative DR approaches in the literature by not aligning based a pairwise comparison of points, i.e. individual $p_{ij}$ and $q_{ij}$. MAP IT exploits combined information represented by marginal probabilities $p_j$ and $q_j$, which are aligned, and which naturally encompass information about multiple pairs of points simultaneously, using unnormalized quantities. This represents a major shift.

## 4   MAP IT'S VISUALIZATIONS

**General comments to the experimental part.** The MAP IT theory has been developed from the viewpoint of divergence and for that reason the t/Gaussian distributions (and perplexity computation) as in t-SNE are chosen (see Appendix C for further considerations on these choices.). The UMAP approach to metrics and weights could have been used, only amounting to design choices. In all MAP IT experiments, a random initialization is used, a perplexity value of 15 is used, and no gain or "trick" are used such as to multiply $\tilde{p}_{jk}$ by some constant in the first (say, 100) iterations (widely done in the t-SNE literature). The common delta-bar-delta rule is used, setting momentum to $0.5$ for the first 100 iterations and then $0.8$. The learning rate is set to 50 always, over 1000 iterations. Results are found to be relatively stable with these choices (see Appendix C for more comments on implementation). For t-SNE the highly optimized Barnes-Hut algorithm is used with its default parameters. For UMAP, the state-of-the-art implementation from the Herzenberg Lab, Stanford, is used with default parameters, thus always initialized by Laplacian eigenmaps. The PacMap author's Python implementation is used[5] with default parameters, thus always initialized by PCA. All methods return a cost function value used to select the best result out of several runs. MAP IT code is available at `https://github.com/SFI-Visual-Intelligence/`.

The focus of this paper is to introduce MAP IT as a fundamentally new way to approach dimensionality reduction and not to extensively scale up MAP IT. For that reason, the focus is on a range of relatively modest sized data sets, to convey the basic properties of MAP IT as a new approach. In Appendix C, some further considerations on potential upscaling of the method based on sampling of forces and computation of (entropy) weights (Eq. (19)) are provided. Visualizations below are best viewed in color. Plots are enlarged and repeated in Appendix C, for the benefit of the reader, where also more details about data sets are given.

---

[5]t-SNE: Barnes-Hut. UMAP: C. Meehan, J. Ebrahimian, W. Moore, and S. Meehan (2022). Uniform Manifold Approximation and Projection (UMAP) (https://www.mathworks.com/matlabcentral/fileexchange/71902). PacMap: https://github.com/YingfanWang/PaCMAP/tree/master.

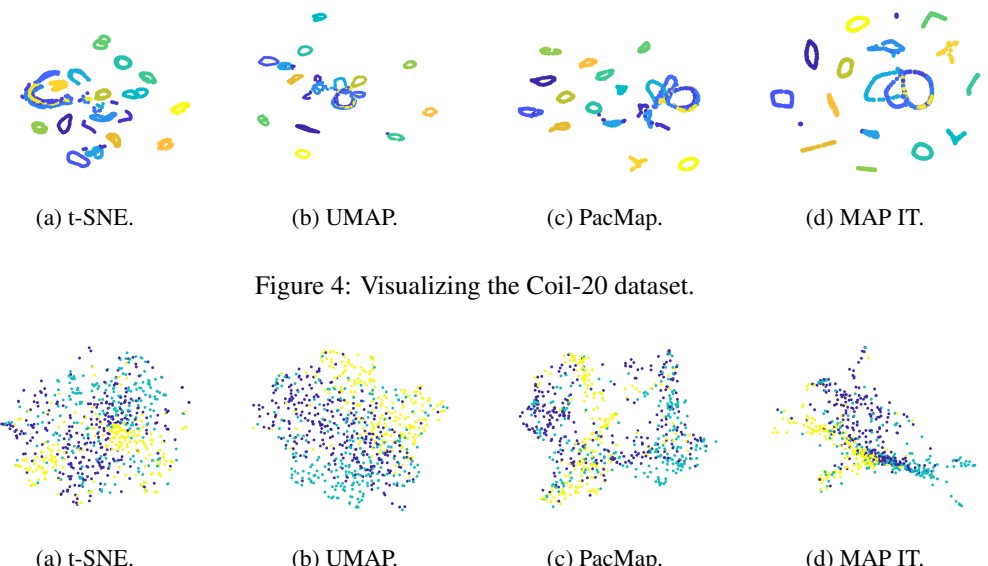

| (a) t-SNE. | (b) UMAP. | (c) PacMap. | (d) MAP IT. |

Figure 4: Visualizing the Coil-20 dataset.

| (a) t-SNE. | (b) UMAP. | (c) PacMap. | (d) MAP IT. |

Figure 5: Visualizing visual concepts.

**MNIST.** A random subset of MNIST consisting of 2000 points is used. Figure 1 has already illustrated that MAP IT, despite not being heavily optimized and by random initialization, produced markedly different representation compared to the alternatives, with a clear class structure and with better separation between classes. Figure 10 in Appendix C shows visualizations using different values $k$ of nearest neighbors. The value of this hyperparameter will influence results. Based on MNIST and other data sets it appears as if a value between 5 and 15 produce reasonable results over a range of data sets. Figure 3 illustrates MAP IT with $k = 10$ for MNIST over different runs. The main structure of the mapped data is always the same showing robustness wrt. initialization. Appendix C provides further quantitative analysis.

**Coil 20.** It can be seen in Figure 4 that MAP IT separates very well the 20 classes, when compared to the alternatives. It may be that MAP IT experiences symmetry breaking to a somewhat larger degree than e.g. PacMap (flattening out ring structures).

**Visual Concepts.** Images corresponding to three different visual concepts are visualized in Fig. (5 by their SIFT (Lowe, 1999) descriptors (1000-dimensional). The visual concepts used are *strawberry* (blue), *lemon* (cyan) and *australian terrier* (yellow). It is evident from all the methods that the concepts are overlapping. t-SNE splits the *australian terrier*. PacMap splits the hugely overlapping groups *strawberry* and *lemon*. UMAP doesn't split groups but indicates little group structure. MAP IT indicates more of a group structure, without splitting any group.

**Newsgroups.** A bag of 100 words is created. Randomly, 10% of the documents are selected, obtaining a total of 1625 documents distributed over the categories `comp`, `rec`, `sci` and `talk`, and a document-word matrix is formed and used to visualize the word distribution, Fig. 6 (see Appendix C for word-clouds). t-SNE and UMAP uniformly spread the words in the plane. PacMap seems to put words in groups roughly corresponding to topics, which may be natural but which is too strict (the word "computer", for instance, would be expected to not be exclusive to e.g. to group `sci`). MAP IT also puts words in groups to some degree, but not exclusively, like PacMap.

**Frey faces.** The 1965 $(28 \times 20)$ "Frey faces" are visualized and shown in Fig. 7. Each image is a frame in a video of the same face varying smoothly over time. t-SNE, UMAP, and PacMap all seem to predominantly pick up on a "smile" versus "no smile" structure in the video. MAP IT creates smaller local structures but these structures seem more spread out in the space in the sense of for instance not placing smiling faces very far from other faces.

**Sampling over non-neighbors.** As mentioned, scaling up MAP IT is not a main focus in this paper. It is however of interest for future work. As an initial experiment, computation of attractive/repulsive

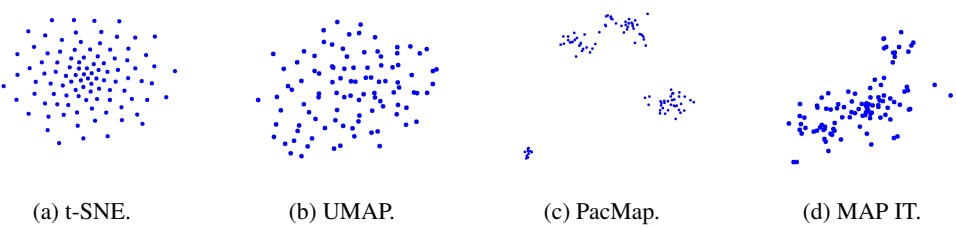

(a) t-SNE.           (b) UMAP.           (c) PacMap.           (d) MAP IT.

Figure 6: Visualizing word-representations from Newsgroups.

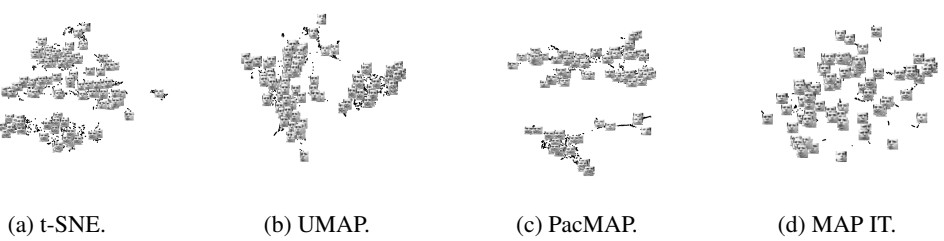

(a) t-SNE.           (b) UMAP.           (c) PacMAP.           (d) MAP IT.

Figure 7: Visualizing the Frey faces.

forces are split on two groups over $k$ neighbors and $n - k$ non-neighbors for each point. The latter group is sampled, using only some multiple of $k$, here $3k$. For $k = 10$ and $n = 2000$, this yields over 98.5 percent reduction in computations. Fig. 8 shows for MNIST that the visualization is basically unchanged (elaborated more in Appendix C).

## 5 CONCLUSIONS

MAP IT takes a fundamentally different approach to visualize representations by aligning distributions over discrete marginal probabilities (node degrees) in the input space versus the target space, thus capturing information in wider local regions. This is contrary to current methods which align based on individual probabilities between pairs of data points (states) only. The MAP IT theory reveals that alignment based on a projective divergence avoids normalization of weights (to obtain true probabilities) entirely, and further reveals a dual viewpoint via continuous densities and kernel smoothing. MAP IT is shown to produce visualizations which capture class structure better than the current state of the art.

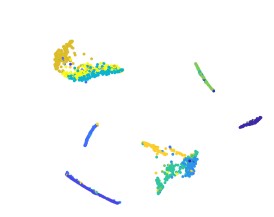

Figure 8: MAP IT with sampling of non-neighbor attractive/repulsive forces.

ACKNOWLEDGMENTS

This work was partially funded by the Research Council of Norway (RCN) grant no. 309439 Centre of Research-based Innovation *Visual Intelligence*, http://visual-intelligence.no, as well as RCN grant no. 303514. The author thanks the anonymous reviewers for helpful comments.

**Reproducibility statement.** To ensure reproducibility, proofs are included in Appendix B and additional results and analysis are included in Appendix C. Code is available at https://github.com/SFI-Visual-Intelligence/.

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

## APPENDIX A: FURTHER PERSPECTIVES ON T-SNE, UMAP, PACMAP, AND VARIANTS

As mentioned in Section 1 of this paper, a large plethora of dimensionality reduction methods exists and an excellent repository for more information is e.g. `https://jlmelville.github.io/smallvis/`. In this paper, following recent literature, the main algorithms are considered to be t-SNE (van der Maaten, 2014), and UMAP (McInnes et al., 2020), and we include the empirically motivated PacMap (Wang et al., 2021). For context, TriMAP (Amid & Warmuth, 2019) and LargeVis (Tang et al., 2016) are discussed below. The t-SNE theory has been outlined in Section 2.

In the paper introducing UMAP, McInnes et al. (2020) argues that t-SNE should be considered the current state-of-the-art at that time, and mentions computational scalability as a main benefit of UMAP versus t-SNE. The main aspect with respect to computational scalability for UMAP versus t-SNE is that the simplical set theory shows that for UMAP normalization over pairwise similarities (probabilities in t-SNE) is not needed, as opposed to t-SNE. This illustrates the importance of the sound theoretical foundation of UMAP. As further described in (McInnes et al., 2020), UMAP's simplical set cross-entropy cost function resembles in several ways the LargeVis (Tang et al., 2016) cost function. LargeVis also avoids normalization in the embedding space, albeit from a more heuristic point of view, but not in the input space where a procedure similar to the one used in Barnes-Hut t-SNE (van der Maaten, 2014) is used. Avoiding normalization in the embedding space

is key to the negative sampling strategy employed in LargeVis and which is key to its computational scalability, also an integral component in the UMAP optimization. LargeVis is not included in the experimental part of this paper to avoid clutter, but is an influential algorithm in the t-SNE family. McInnes et al. (2020) and Wang et al. (2021), for instance, have both extensive comparative experiments sections also involving LargeVis.

A motivation for Wang et al. (2021) is to discuss preservation of local structure versus global structure. They propose a heuristic method, PacMap, which is intended to strike a balance between TriMap (Amid & Warmuth, 2019) (better at preserving global structure) and t-SNE/UMAP (local structure). TriMap is is a triplet loss-based method. Wang et al. (2021) argues that TriMap is the first successful triplet constraint method (as opposed to (Hadsell et al., 2006; van der Maaten & Weinberger, 2012; Wilber et al., 2015)) but claims that without PCA initialization "TriMap's global structure is ruined". PacMap is based on a study of the principles behind attractive and repulsive forces and finds that forces should be exerted on further points and sets up a heuristically designed procedure for treating near pairs, mid-near pairs, and non-neighbors.

Understanding t-SNE versus UMAP, in particular, from a theoretical perspective, has gained interest in the recent years. Damrich & Hamprecht (2021) studies the interplay between attractive and repulsive forces in UMAP in detail and comes to the conclusion that UMAP is actually not exactly optimizing the cost function put forth in (McInnes et al., 2020). Böhm et al. (2022) studies the whole attraction-repulsion spectrum and find cases where UMAP may diverge. Damrich et al. (2023) provides an entirely new viewpoint relating a fixed normalization constant in UMAP to a smaller learned normalization constant in t-SNE when considering negative sampling and noise contrastive learning. Draganov et al. (2023) argues that the normalization aspect is basically the key difference between t-SNE and UMAP, and suggests a way to toggle between the two approaches.

Kobak & Linderman (2021) show that UMAP's initialization (Laplacian eigenmaps (Belkin & Niyogi, 2003)) is very important for UMAP's results and claims that t-SNE can be improved by a similar initialization. Wang et al. (2021) also studies initialization, and claim that both UMAP but also TriMap are very dependent on initialization. Several papers have revealed close connections between t-SNE and Laplacian eigenmaps (Carreira-Perpinan, 2010; Linderman & Steinerberger, 2019; Böhm et al., 2022). Some further comments on the relationship between t-SNE and Laplacian eigenmaps are provided in Appendix B. Draganov et al. (2023) argues that the normalization aspect is basically the key difference between t-SNE and UMAP, and suggests a way to toggle between the two approaches. Other highly influential papers provide important insight (Kobak & Berens, 2019; Kobak et al., 2020).

In this paper, sampling is also demonstrated (Figure 8, and Appendix C), sharing some similarity to the sampling strategies invoked in e.g. LargeVis and UMAP.

It should also be mentioned that dimensionality reduction methods inspired by the t-SNE approach by alternative divergences to the Kullback-Leibler over joint pairwise probabilities have been studied to some degree (Bunte et al., 2012; Naryan et al., 2015; Huang et al., 2022). However, these works have not discussed projective properties of divergence measures and have not contributed to understanding the aspect of normalization.

## APPENDIX B: PROPOSITIONS WITH PROOFS AND COMMENTS

For the benefit of the reader, the well-known Kullback-Leibler-based t-SNE relation from the main paper, which van der Maaten & Hinton (2008) builds on, is proved:

$$\arg\min_{\boldsymbol{z}_1,...,\boldsymbol{z}_n \in \mathbb{R}_d} KL(P||Q) = \arg\min_{\boldsymbol{z}_1,...,\boldsymbol{z}_n \in \mathbb{R}_d} \sum_{i,j} p_{ij} \log \frac{p_{ij}}{q_{ij}} = \arg\min_{\boldsymbol{z}_1,...,\boldsymbol{z}_n \in \mathbb{R}_d} \sum_{i,j} p_{ij} \log q_{ij}. \quad (20)$$

*Proof.*

$$\underset{\boldsymbol{z}_1,\dots,\boldsymbol{z}_n\in\mathbb{R}_d}{\arg\min}\ KL(P||Q) = \underset{\boldsymbol{z}_1,\dots,\boldsymbol{z}_n\in\mathbb{R}_d}{\arg\min}\ \sum_{i,j} p_{ij}\log\frac{p_{ij}}{q_{ij}} \tag{21}$$

$$= \underset{\boldsymbol{z}_1,\dots,\boldsymbol{z}_n\in\mathbb{R}_d}{\arg\min}\ \underbrace{\sum_{i,j} p_{ij}\log p_{ij}}_{\text{constant}} - \sum_{i,j} p_{ij}\log q_{ij}. \tag{22}$$

$$= \underset{\boldsymbol{z}_1,\dots,\boldsymbol{z}_n\in\mathbb{R}_d}{\arg\min}\ -\sum_{i,j} p_{ij}\log q_{ij}. \tag{23}$$

$\square$

**Proposition 1.** *[Minimizing $KL(P||Q)$ and the role of normalization]. Let $q_{ij} = \tilde{q}_{ij}/Z_q$ for some choice of $\tilde{q}_{ij}$ where $Z_q = \sum\limits_{n,m} \tilde{q}_{nm}$. Then*

$$\underset{\boldsymbol{z}_1,\dots,\boldsymbol{z}_n\in\mathbb{R}_d}{\arg\min}\ KL(P||Q) = \underset{\boldsymbol{z}_1,\dots,\boldsymbol{z}_n\in\mathbb{R}_d}{\arg\min}\ -\sum_{i,j} p_{ij}\log\tilde{q}_{ij} + \log\sum_{n,m}\tilde{q}_{nm}. \tag{24}$$

*Proof.*

$$\underset{\boldsymbol{z}_1,\dots,\boldsymbol{z}_n\in\mathbb{R}_d}{\arg\min}\ KL(P||Q) = \underset{\boldsymbol{z}_1,\dots,\boldsymbol{z}_n\in\mathbb{R}_d}{\arg\min}\ -\sum_{i,j} p_{ij}\log q_{ij} \tag{25}$$

$$= \underset{\boldsymbol{z}_1,\dots,\boldsymbol{z}_n\in\mathbb{R}_d}{\arg\min}\ -\sum_{i,j} p_{ij}\log\frac{\tilde{q}_{ij}}{\sum\limits_{n,m}\tilde{q}_{nm}} \tag{26}$$

$$= \underset{\boldsymbol{z}_1,\dots,\boldsymbol{z}_n\in\mathbb{R}_d}{\arg\min}\ -\left(\sum_{i,j} p_{ij}\log\tilde{q}_{ij} - \underbrace{\sum_{i,j} p_{ij}}_{=1}\log\sum_{n,m}\tilde{q}_{nm}\right) \tag{27}$$

$$= \underset{\boldsymbol{z}_1,\dots,\boldsymbol{z}_n\in\mathbb{R}_d}{\arg\min}\ -\sum_{i,j} p_{ij}\log\tilde{q}_{ij} + \log\sum_{n,m}\tilde{q}_{nm}. \tag{28}$$

$\square$

**Comment to Proposition 1.** Deriving the t-SNE cost function without first expressing $q_{ij}$[6] in a particular form (t-distribution (Cauchy distribution) or Gaussian) shows a general property of the cost function. The t-SNE cost function is by the above result expressed by two terms where the first term relates the normalized $p_{ij}$ to unnormalized $\tilde{q}_{ij}$. The second term only involves the unnormalized $\tilde{q}_{ij}$.

From this expression, an interesting aspect of the t-SNE cost function is an intrinsic connection to Laplacian eigenmaps (Belkin & Niyogi, 2003). Discovery of connections between t-SNE and Laplacian eigenmaps can be traced back to Carreira-Perpinan (2010). More recently, Böhm et al. (2022) performed a closer examination of this relationship in terms of eigenvectors of normalized and unnormalized Laplacian matrices. Linderman & Steinerberger (2019) studied convergence of clustering with t-SNE from the viewpoint of early exaggeration (an often used optimization "trick") and identified a regime where t-SNE behaves like Laplacian-based spectral clustering.

In Laplacian eigenmaps, the aim is to find a low-dimensional embedding $\boldsymbol{z}_1,\dots,\boldsymbol{z}_n \in \mathbb{R}_d$ from $\boldsymbol{x}_1,\dots,\boldsymbol{x}_n \in \mathbb{R}_D$. It is assumed that some similarity measure can be defined in the input space,

---

[6]In the original SNE paper (Hinton & Roweis, 2002) the Gaussian distribution was used $q_{ij} = \exp(-\kappa||\boldsymbol{z}_i - \boldsymbol{z}_j||^2)/\sum_{n,m}\exp(-\kappa||\boldsymbol{z}_n - \boldsymbol{z}_m||^2)$. The argument in (van der Maaten & Hinton, 2008) was that the t-distribution helps mitigate the so-called crowding problem. It is possible to formulate the joint probabilities as functions of other distance functions than the Euclidean or in terms of similarity measures.

pairwise over $x_i$ and $x_j$, which can be denoted $w_{ij}$. The Laplacian cost function is essentially $\sum_{i,j} w_{ij} ||z_i - z_j||^2$ to be minimized over $z_1, \ldots, z_n \in \mathbb{R}_d$, however given orthogonality constraints on $z_1, \ldots, z_n \in \mathbb{R}_d$ to avoid trivial minima. Since for t-SNE, $\tilde{q}_{ij}$ is a function of $||z_i - z_j||^2$ for both the t-distribution (Cauchy distribution) and the Gaussian, there is a close link between t-SNE and Laplacian eigenmaps. For instance, for $\tilde{q}_{ij} = [1 + ||z_i - z_j||^2]^{-1}$, the first term becomes $\sum_{i,j} w_{ij} \log ||z_i - z_j||^2$ and for the Gaussian the corresponding term becomes $\sum_{i,j} w_{ij} ||z_i - z_j||^2$ up to a proportionality constant (this was also pointed out in (Trosten et al., 2023)). By letting $w_{ij} = p_{ij}$ the link becomes obvious. In t-SNE, there are no orthogonality constraints on $z_1, \ldots, z_n \in \mathbb{R}_d$. Instead, trivial solutions are avoided by the constraint posed by $\sum_{n,m} \tilde{q}_{nm}$.

Recently, Cai & Ma (2022) provided an elaborate spectral analysis concluding that t-SNE is connected to Laplacian eigenmaps. By the above analysis, and from e.g. (Böhm et al., 2022), this is evident from the form of the t-SNE cost function itself.

**Proposition 2.** *[Gradient of $KL(P||Q)$]*

$$\frac{\partial}{\partial z_i} KL(P||Q) = -4 \sum_j (p_{ij} - q_{ij}) \tilde{q}_{ij} (z_j - z_i). \tag{29}$$

*Proof.*

$$\frac{\partial}{\partial z_i} KL(P||Q) = \frac{\partial}{\partial z_i} \underbrace{- \sum_{i,j} p_{ij} \log \tilde{q}_{ij} + \log \sum_{n,m} \tilde{q}_{nm}}_{\stackrel{\text{def}}{=} C}. \tag{30}$$

The following derivation resembles (van der Maaten & Hinton, 2008). Note that if $z_i$ changes, the only pairwise distances that change are $d_{ij}$ and $d_{ji}$ where $d_{ij} = ||z_i - z_j||$. Hence, the gradient of the cost function $C$ with respect to $z_i$ is given by

$$\frac{\partial}{\partial z_i} C = \sum_j \left( \frac{\partial C}{\partial d_{ij}} + \frac{\partial C}{\partial d_{ji}} \right) (z_i - z_j) = 2 \sum_j \frac{\partial C}{\partial d_{ij}} (z_i - z_j). \tag{31}$$

Furthermore

$$\frac{\partial C}{\partial d_{ij}} = - \sum_{k,l} p_{kl} \frac{\partial}{\partial d_{ij}} \log \tilde{q}_{kl} + \frac{\partial}{\partial d_{ij}} \log \sum_{n,m} \tilde{q}_{nm} \tag{32}$$

$$= - \sum_{k,l} p_{kl} \frac{1}{\tilde{q}_{kl}} \frac{\partial}{\partial d_{ij}} \tilde{q}_{kl} + \frac{1}{\sum_{n,m} \tilde{q}_{nm}} \sum_{n,m} \frac{\partial}{\partial d_{ij}} \tilde{q}_{nm}. \tag{33}$$

The gradient is only non-zero for $k = i$, $l = j$ and for $n = i$, $m = j$, yielding

$$\frac{\partial C}{\partial d_{ij}} = -p_{ij} \frac{1}{\tilde{q}_{ij}} \frac{\partial}{\partial d_{ij}} \tilde{q}_{ij} + \frac{1}{\sum_{n,m} \tilde{q}_{nm}} \frac{\partial}{\partial d_{ij}} \tilde{q}_{ij}. \tag{34}$$

Note that for $\tilde{q}_{ij} = \left[ \frac{1}{1 + ||z_i - z_j||^2} \right]$, which is the t-distribution, or alternatively for $\tilde{q}_{ij} = \exp(-\kappa ||z_i - z_j||^2)$, the Gaussian distribution, we have

$$\frac{\partial}{\partial d_{ij}} \tilde{q}_{ij} = 2 \tilde{q}_{ij}^2. \tag{35}$$

Hence,

$$\frac{\partial C}{\partial d_{ij}} = 2 \left( -p_{ij} \tilde{q}_{ij} + \frac{1}{\sum_{n,m} \tilde{q}_{nm}} \tilde{q}_{ij}^2 \right) = 2 (-p_{ij} + q_{ij}) \tilde{q}_{ij}, \tag{36}$$

since $q_{ij} = \frac{\tilde{q}_{ij}}{\sum\limits_{n,m} \tilde{q}_{nm}}$. Finally, this yields

$$\frac{\partial}{\partial \boldsymbol{z}_i} C = \frac{\partial}{\partial \boldsymbol{z}_i} KL(P||Q) = -4 \sum_j (p_{ij} - q_{ij}) \tilde{q}_{ij}(\boldsymbol{z}_j - \boldsymbol{z}_i). \tag{37}$$

$\square$

**Proposition 3.** *[CS divergence is projective]. Let $p_j = \tilde{p}_j/Z_p$, $Z_p = \sum_{j'} \tilde{p}_{j'}$, and $q_j = \tilde{q}_j/Z_q$, $Z_q = \sum_{j'} \tilde{q}_{j'}$. Then*

$$CS(P_m||Q_m) = CS(\tilde{P}_m||\tilde{Q}_m). \tag{38}$$

*Proof.*

$$CS(P_m||Q_m) = -\log \frac{\sum\limits_j p_j q_j}{\left(\sum\limits_j p_j^2\right)^{\frac{1}{2}} \left(\sum\limits_j q_j^2\right)^{\frac{1}{2}}} \tag{39}$$

$$= -\log \frac{\sum\limits_j \frac{\tilde{p}_j}{Z_p} \frac{\tilde{q}_j}{Z_q}}{\left(\sum\limits_j \left(\frac{\tilde{p}_j}{Z_p}\right)^2\right)^{\frac{1}{2}} \left(\sum\limits_j \left(\frac{\tilde{q}_j}{Z_q}\right)^2\right)^{\frac{1}{2}}} \tag{40}$$

$$= -\log \frac{\sum\limits_j \tilde{p}_j \tilde{q}_j}{\left(\sum\limits_j \tilde{p}_j^2\right)^{\frac{1}{2}} \left(\sum\limits_j \tilde{q}_j^2\right)^{\frac{1}{2}}} \tag{41}$$

$$= CS(\tilde{P}_m||\tilde{Q}_m). \tag{42}$$

$\square$

**Proposition 4.** *[Minimizing $CS(P_m||Q_m)$ with respect to $\boldsymbol{z}_1, \ldots, \boldsymbol{z}_n \in \mathbb{R}_d$]*

$$\underset{\boldsymbol{z}_1,\ldots,\boldsymbol{z}_n \in \mathbb{R}_d}{\arg\min} CS(\tilde{P}_m||\tilde{Q}_m) = \underset{\boldsymbol{z}_1,\ldots,\boldsymbol{z}_n \in \mathbb{R}_d}{\arg\min} -\log \sum_j \tilde{p}_j \tilde{q}_j + \frac{1}{2}\log \sum_j \tilde{q}_j^2. \tag{43}$$

*Proof.*

$$\underset{\boldsymbol{z}_1,\ldots,\boldsymbol{z}_n \in \mathbb{R}_d}{\arg\min} CS(P_m||Q_m) \tag{44}$$

$$= \underset{\boldsymbol{z}_1,\ldots,\boldsymbol{z}_n \in \mathbb{R}_d}{\arg\min} -\log \frac{\sum\limits_j \tilde{p}_j \tilde{q}_j}{\underbrace{\left(\sum\limits_j \tilde{p}_j^2\right)^{\frac{1}{2}}}_{\text{independent of } \boldsymbol{z}} \left(\sum\limits_j \tilde{q}_j^2\right)^{\frac{1}{2}}} \tag{45}$$

$$= \underset{\boldsymbol{z}_1,\ldots,\boldsymbol{z}_n \in \mathbb{R}_d}{\arg\min} -\log \sum_j \tilde{p}_j \tilde{q}_j + \frac{1}{2}\log \sum_j \tilde{q}_j^2. \tag{46}$$

$$\tag{47}$$

$\square$

**Proposition 5.** *[Gradient of $CS(P_m||Q_m)$]*

$$\frac{\partial}{\partial \boldsymbol{z}_i} CS(P_m||Q_m) = -4 \sum_j \left[ \frac{\tilde{p}_j}{\sum\limits_{j'} \tilde{p}_{j'} \tilde{q}_{j'}} - \frac{\tilde{q}_j}{\sum\limits_{j'} \tilde{q}_{j'}^2} \right] \tilde{q}_{ij}^2 (\boldsymbol{z}_j - \boldsymbol{z}_i). \tag{48}$$

*Proof.*

$$\underset{\boldsymbol{z}_1,\dots,\boldsymbol{z}_n \in \mathbb{R}_d}{\arg\min} \; CS(P_m||Q_m) = \underset{\boldsymbol{z}_1,\dots,\boldsymbol{z}_n \in \mathbb{R}_d}{\arg\min} \; -\log \sum_j \tilde{p}_j \tilde{q}_j + \frac{1}{2} \log \sum_j \tilde{q}_j^2 \tag{49}$$

$$\tag{50}$$

There are several ways to proceed. Here, it is chosen to start by expressing $CS(P_m||Q_m)$ explicitly into cross-product terms $\tilde{p}_{jk'}\tilde{q}_{jk}$. For convenience, the derivation is split into two parts.

$$\frac{\partial}{\partial \boldsymbol{z}_i} -\log \sum_j \tilde{p}_j \tilde{q}_j = -\frac{1}{\sum\limits_{j'} \tilde{p}_{j'}\tilde{q}_{j'}} \frac{\partial}{\partial \boldsymbol{z}_i} \sum_j \tilde{p}_j \tilde{q}_j = -\frac{1}{\sum\limits_{j'} \tilde{p}_{j'}\tilde{q}_{j'}} \frac{\partial}{\partial \boldsymbol{z}_i} \sum_j \sum_{k',k} \tilde{p}_{jk'}\tilde{q}_{jk}. \tag{51}$$

Look first at the case $j \neq i$. Then $\frac{\partial}{\partial \boldsymbol{z}_i} \sum\limits_{k',k} \tilde{p}_{jk'}\tilde{q}_{jk}$ will have non-zero terms for $k = i$, hence

$$\frac{\partial}{\partial \boldsymbol{z}_i} \sum_{k',k} \tilde{p}_{jk'}\tilde{q}_{jk} = \sum_{k'} \tilde{p}_{jk'} \frac{\partial}{\partial \boldsymbol{z}_i} \tilde{q}_{ji} = \tilde{p}_j \frac{\partial}{\partial \boldsymbol{z}_i} \tilde{q}_{ji}. \tag{52}$$

For $j = i$,

$$\frac{\partial}{\partial \boldsymbol{z}_i} \sum_{k',k} \tilde{p}_{ik'}\tilde{q}_{ik} = \sum_k \left( \sum_{k'} \tilde{p}_{ik'} \right) \frac{\partial}{\partial \boldsymbol{z}_i} \tilde{q}_{ik} = \sum_k \tilde{p}_k \frac{\partial}{\partial \boldsymbol{z}_i} \tilde{q}_{ik}. \tag{53}$$

Hence,

$$\frac{\partial}{\partial \boldsymbol{z}_i} \sum_j \sum_{k',k} \tilde{p}_{jk'}\tilde{q}_{jk} = \sum_{j,j \neq i} 2\tilde{p}_j \frac{\partial}{\partial \boldsymbol{z}_i} \tilde{q}_{ij}. \tag{54}$$

Note that for $\tilde{q}_{ij} = \left[ \frac{1}{1+||\boldsymbol{z}_i - \boldsymbol{z}_j||^2} \right]$, which is the t-distribution, or alternatively for $\tilde{q}_{ij} = \exp(-\kappa||\boldsymbol{z}_i - \boldsymbol{z}_j||^2)$, the Gaussian distribution, we have

$$\frac{\partial}{\partial \boldsymbol{z}_i} \tilde{q}_{ij} = 2\tilde{q}_{ij}^2 (\boldsymbol{z}_j - \boldsymbol{z}_i) \tag{55}$$

Thus

$$\frac{\partial}{\partial \boldsymbol{z}_i} -\log \sum_j \tilde{p}_j \tilde{q}_j = -4 \frac{1}{\sum\limits_{j'} \tilde{p}_{j'}\tilde{q}_{j'}} \sum_{j,j \neq i} \tilde{p}_j \tilde{q}_{ij}^2 (\boldsymbol{z}_j - \boldsymbol{z}_i). \tag{56}$$

Alternatively,

$$\frac{\partial}{\partial \boldsymbol{z}_i} -\log \sum_j \tilde{p}_j \tilde{q}_j = -\frac{1}{\sum\limits_{j'} \tilde{p}_{j'}\tilde{q}_{j'}} \frac{\partial}{\partial \boldsymbol{z}_i} \sum_i \tilde{p}_i \tilde{q}_i = -\frac{1}{\sum\limits_{j'} \tilde{p}_{j'}\tilde{q}_{j'}} \sum_j \tilde{p}_j \frac{\partial}{\partial \boldsymbol{z}_i} \tilde{q}_j \tag{57}$$

and then work with $\tilde{q}_j = \sum\limits_{k'} \tilde{q}_{jk'}$. For the second part, consider

$$\frac{\partial}{\partial \boldsymbol{z}_i} \frac{1}{2} \log \sum_j \tilde{q}_j^2 = \frac{1}{2} \frac{1}{\sum\limits_{j'} \tilde{q}_{j'}^2} \frac{\partial}{\partial \boldsymbol{z}_i} \sum_i \tilde{q}_i^2 = \frac{1}{2} \frac{1}{\sum\limits_{j'} \tilde{q}_{j'}^2} \frac{\partial}{\partial \boldsymbol{z}_i} \sum_j \sum_{k',k} \tilde{q}_{jk'}\tilde{q}_{jk}. \tag{58}$$

and work in a similar fashion as above from there, or express

$$\frac{\partial}{\partial \boldsymbol{z}_i} \frac{1}{2} \log \sum_j \tilde{q}_j^2 = \frac{1}{2} \frac{1}{\sum_{j'} \tilde{q}_{j'}^2} \frac{\partial}{\partial \boldsymbol{z}_i} \sum_i \tilde{q}_i^2 = \frac{1}{2} \frac{1}{\sum_{j'} \tilde{q}_{j'}^2} \sum_j 2\tilde{q}_j \frac{\partial}{\partial \boldsymbol{z}_i} \tilde{q}_j \tag{59}$$

and insert $\tilde{q}_j = \sum_{k'} \tilde{q}_{jk'}$. This gives

$$\frac{\partial}{\partial \boldsymbol{z}_i} \frac{1}{2} \log \sum_j \tilde{q}_j^2 = 4 \frac{1}{\sum_{j'} \tilde{q}_{j'}^2} \sum_{j,j\neq i} \tilde{q}_j \tilde{q}_{ij}^2 (\boldsymbol{z}_j - \boldsymbol{z}_i). \tag{60}$$

Hence, when taken together:

$$\frac{\partial}{\partial \boldsymbol{z}_i} CS(P_m \| Q_m) = -4 \sum_{j,j\neq i} \left[ \frac{\tilde{p}_j}{\sum_{j'} \tilde{p}_{j'} \tilde{q}_{j'}} - \frac{\tilde{q}_j}{\sum_{j'} \tilde{q}_{j'}^2} \right] \tilde{q}_{ij}^2 (\boldsymbol{z}_j - \boldsymbol{z}_i). \tag{61}$$

$\square$

**Proposition 6.** *[The* map *CS divergence is projective]. Let* $\boldsymbol{z} = m(\boldsymbol{x}) + \varepsilon$ *and assume* $p(\boldsymbol{x}) = \tilde{p}(\boldsymbol{x})/Z_p$ *and* $q(\boldsymbol{z}) = \tilde{q}(\boldsymbol{z})/Z_q$ *where* $\tilde{p}(\boldsymbol{x})$ *and* $\tilde{q}(\boldsymbol{z})$ *are unnormalized with* $Z_p = \int \tilde{p}(\boldsymbol{x})d\boldsymbol{x}$ *and* $Z_q = \int \tilde{q}(\boldsymbol{z})d\boldsymbol{z}$ *as the respective normalization constants. Then*

$$CS(p(\boldsymbol{x})\|q(\boldsymbol{z})) = CS(\tilde{p}(\boldsymbol{x})\|\tilde{q}(\boldsymbol{z})). \tag{62}$$

*Proof.*

$$CS(p(\boldsymbol{x})\|q(\boldsymbol{z})) \tag{63}$$

$$= -\log \frac{\int \int \frac{\tilde{p}(\boldsymbol{x})}{Z_p} \frac{\tilde{q}(\boldsymbol{z})}{Z_q} f(\boldsymbol{x}, \boldsymbol{z}) d\boldsymbol{x} d\boldsymbol{z}}{\left( \int \int \left( \frac{\tilde{p}(\boldsymbol{x})}{Z_p} \right)^2 f(\boldsymbol{x}, \boldsymbol{z}) d\boldsymbol{x} d\boldsymbol{z} \right)^{\frac{1}{2}} \left( \int \int \left( \frac{\tilde{q}(\boldsymbol{z})}{Z_q} \right)^2 f(\boldsymbol{x}, \boldsymbol{z}) d\boldsymbol{x} d\boldsymbol{z} \right)^{\frac{1}{2}}} \tag{64}$$

$$= -\log \frac{\frac{1}{Z_p Z_q} \int \int \tilde{p}(\boldsymbol{x})\tilde{q}(\boldsymbol{z}) f(\boldsymbol{x}, \boldsymbol{z}) d\boldsymbol{x} d\boldsymbol{z}}{\left( \frac{1}{Z_p^2} \int \int \tilde{p}^2(\boldsymbol{x}) f(\boldsymbol{x}, \boldsymbol{z}) d\boldsymbol{x} d\boldsymbol{z} \right)^{\frac{1}{2}} \left( \frac{1}{Z_q^2} \int \int \tilde{q}^2(\boldsymbol{z}) f(\boldsymbol{x}, \boldsymbol{z}) d\boldsymbol{x} d\boldsymbol{z} \right)^{\frac{1}{2}}} \tag{65}$$

$$= -\log \frac{\int \int \tilde{p}(\boldsymbol{x})\tilde{q}(\boldsymbol{z}) f(\boldsymbol{x}, \boldsymbol{z}) d\boldsymbol{x} d\boldsymbol{z}}{\left( \int \int \tilde{p}^2(\boldsymbol{x}) f(\boldsymbol{x}, \boldsymbol{z}) d\boldsymbol{x} d\boldsymbol{z} \right)^{\frac{1}{2}} \left( \int \int \tilde{q}^2(\boldsymbol{z}) f(\boldsymbol{x}, \boldsymbol{z}) d\boldsymbol{x} d\boldsymbol{z} \right)^{\frac{1}{2}}} \tag{66}$$

$$= CS(\tilde{p}(\boldsymbol{x})\|\tilde{q}(\boldsymbol{z})). \tag{67}$$

$\square$

**Proposition 7.** *[Empirical* map *CS divergence]. Let a sample* $\boldsymbol{x}_1, \ldots, \boldsymbol{x}_n \in \mathbb{R}_D$ *be given and assume the mapping* $\boldsymbol{z} = m(\boldsymbol{x}) + \varepsilon$. *Then, an empirical* map *CS divergence is given by*

$$\widehat{CS}(p(\boldsymbol{x})\|q(\boldsymbol{z})) = -\log \frac{\sum_j \tilde{p}(\boldsymbol{x}_j)\tilde{q}(\boldsymbol{z}_j)}{\left( \sum_j \tilde{p}^2(\boldsymbol{x}_j) \right)^{\frac{1}{2}} \left( \sum_j \tilde{q}^2(\boldsymbol{z}_j) \right)^{\frac{1}{2}}}. \tag{68}$$

*Proof.* We have $CS(p(\boldsymbol{x})\|q(\boldsymbol{z})) = CS(\tilde{p}(\boldsymbol{x})\|\tilde{q}(\boldsymbol{z}))$. Thus

$$CS(\tilde{p}(\boldsymbol{x})\|\tilde{q}(\boldsymbol{z})) = -\log \frac{\int \int \tilde{p}(\boldsymbol{x})\tilde{q}(\boldsymbol{z}) f(\boldsymbol{x}, \boldsymbol{z}) d\boldsymbol{x} d\boldsymbol{z}}{\left( \int \int \tilde{p}^2(\boldsymbol{x}) f(\boldsymbol{x}, \boldsymbol{z}) d\boldsymbol{x} d\boldsymbol{z} \right)^{\frac{1}{2}} \left( \int \int \tilde{q}^2(\boldsymbol{z}) f(\boldsymbol{x}, \boldsymbol{z}) d\boldsymbol{x} d\boldsymbol{z} \right)^{\frac{1}{2}}}. \tag{69}$$

It suffices to look at the numerator. Since $\int \int \tilde{p}(\boldsymbol{x}) \tilde{q}(\boldsymbol{z}) f(\boldsymbol{x}, \boldsymbol{z}) d\boldsymbol{x} d\boldsymbol{z} = E_{\mathcal{X} \times \mathcal{Z}} [\tilde{p}(\boldsymbol{x}) \tilde{q}(\boldsymbol{z})]$ and we have

$$E_{\mathcal{X} \times \mathcal{Z}} \widehat{[\tilde{p}(\boldsymbol{x})} \tilde{q}(\boldsymbol{z})] = \sum_j \tilde{p}(\boldsymbol{x}_j) \tilde{q}(\boldsymbol{z}_j), \tag{70}$$

we have

$$\widehat{CS}(p(\boldsymbol{x}) \| q(\boldsymbol{z})) = -\log \frac{\sum\limits_j \tilde{p}(\boldsymbol{x}_j) \tilde{q}(\boldsymbol{z}_j)}{\left(\sum\limits_j \tilde{p}^2(\boldsymbol{x}_j)\right)^{\frac{1}{2}} \left(\sum\limits_j \tilde{q}^2(\boldsymbol{z}_j)\right)^{\frac{1}{2}}}. \tag{71}$$

$\square$

**Proposition 8.** *[Minimizing $\widehat{CS}(p(\boldsymbol{x}) \| q(\boldsymbol{z}))$ with respect to $\boldsymbol{z}_1, \ldots, \boldsymbol{z}_n \in \mathbb{R}_d$]*

$$\underset{\boldsymbol{z}_1, \ldots, \boldsymbol{z}_n \in \mathbb{R}_d}{\arg\min} \widehat{CS}(p(\boldsymbol{x}) \| q(\boldsymbol{z})) = \underset{\boldsymbol{z}_1, \ldots, \boldsymbol{z}_n \in \mathbb{R}_d}{\arg\min} -\log \sum_j \tilde{p}(\boldsymbol{x}_j) \tilde{q}(\boldsymbol{z}_j) + \frac{1}{2} \log \sum_j \tilde{q}^2(\boldsymbol{z}_j). \tag{72}$$

*Proof.*

$$\underset{\boldsymbol{z}_1, \ldots, \boldsymbol{z}_n \in \mathbb{R}_d}{\arg\min} CS(p(\boldsymbol{x}) \| q(\boldsymbol{z})) \tag{73}$$

$$= \underset{\boldsymbol{z}_1, \ldots, \boldsymbol{z}_n \in \mathbb{R}_d}{\arg\min} -\log \frac{\sum\limits_j \tilde{p}(\boldsymbol{x}_j) \tilde{q}(\boldsymbol{z}_j)}{\underbrace{\left(\sum\limits_j \tilde{p}^2(\boldsymbol{x}_j)\right)^{\frac{1}{2}}}_{\text{independent of } \boldsymbol{z}} \left(\sum\limits_j \tilde{q}^2(\boldsymbol{z}_j)\right)^{\frac{1}{2}}} \tag{74}$$

$$= \underset{\boldsymbol{z}_1, \ldots, \boldsymbol{z}_n \in \mathbb{R}_d}{\arg\min} -\log \sum_j \tilde{p}(\boldsymbol{x}) \tilde{q}(\boldsymbol{z}_j) + \frac{1}{2} \log \sum_j \tilde{q}^2(\boldsymbol{z}_j). \tag{75}$$

$$\tag{76}$$

$\square$

**Proposition 9.** *[Gradient of $\widehat{CS}(p(\boldsymbol{x}) \| q(\boldsymbol{z}))$]*

$$\frac{\partial}{\partial \boldsymbol{z}_i} \widehat{CS}(p(\boldsymbol{x}) \| q(\boldsymbol{z})) = -\sum_j \left[ \frac{\tilde{p}(\boldsymbol{x}_j)}{\sum\limits_{j'} \tilde{p}(\boldsymbol{x}_{j'}) \tilde{q}(\boldsymbol{z}_{j'})} - \frac{\tilde{q}(\boldsymbol{z}_j)}{\sum\limits_{j'} \tilde{q}^2(\boldsymbol{z}_{j'})} \right] \frac{\partial}{\partial \boldsymbol{z}_i} \tilde{q}(\boldsymbol{z}_j). \tag{77}$$

*Proof.*

$$\underset{\boldsymbol{z}_1, \ldots, \boldsymbol{z}_n \in \mathbb{R}_d}{\arg\min} CS(p(\boldsymbol{x}) \| q(\boldsymbol{z})) = \underset{\boldsymbol{z}_1, \ldots, \boldsymbol{z}_n \in \mathbb{R}_d}{\arg\min} -\log \sum_j \tilde{p}(\boldsymbol{x}_j) \tilde{q}(\boldsymbol{z}_j) + \frac{1}{2} \log \sum_j \tilde{q}^2(\boldsymbol{z}_j). \tag{78}$$

The derivation is split into two parts. First,

$$\frac{\partial}{\partial \boldsymbol{z}_i} -\log \sum_j \tilde{p}(\boldsymbol{x}_j) \tilde{q}(\boldsymbol{z}_j) = -\frac{1}{\sum\limits_{j'} \tilde{p}(\boldsymbol{x}_{j'}) \tilde{q}(\boldsymbol{z}_{j'})} \sum_j \tilde{p}(\boldsymbol{x}_j) \frac{\partial}{\partial \boldsymbol{z}_i} \tilde{q}(\boldsymbol{z}_j). \tag{79}$$

Second,

$$\frac{\partial}{\partial \boldsymbol{z}_i} \frac{1}{2} \log \sum_j \tilde{q}^2(\boldsymbol{z}_j) = \frac{1}{2} \frac{1}{\sum\limits_{j'} \tilde{q}^2(\boldsymbol{z}_{j'})} \sum_j \frac{\partial}{\partial \boldsymbol{z}_i} \tilde{q}^2(\boldsymbol{z}_j) = \frac{1}{2} \frac{1}{\sum\limits_{j'} \tilde{q}^2(\boldsymbol{z}_{j'})} \sum_j 2\tilde{q}(\boldsymbol{z}_j) \frac{\partial}{\partial \boldsymbol{z}_i} \tilde{q}(\boldsymbol{z}_j). \tag{80}$$

Taken together, thus

$$\frac{\partial}{\partial \boldsymbol{z}_i} \widehat{CS}(p(\boldsymbol{x})||q(\boldsymbol{z})) = -\sum_j \left[ \frac{\tilde{p}(\boldsymbol{x}_j)}{\sum_{j'} \tilde{p}(\boldsymbol{x}_{j'})\tilde{q}(\boldsymbol{z}_{j'})} - \frac{\tilde{q}(\boldsymbol{z}_j)}{\sum_{j'} \tilde{q}^2(\boldsymbol{z}_{j'})} \right] \frac{\partial}{\partial \boldsymbol{z}_i} \tilde{q}(\boldsymbol{z}_j). \tag{81}$$

$\square$

**Proposition 10.** *[Gradient of $\widehat{CS}(p(\boldsymbol{x})||q(\boldsymbol{z}))$ with kernel smoothing]. Let $\hat{\tilde{q}}(\boldsymbol{z}_j) = \sum_k \kappa_z(\boldsymbol{z}_j - \boldsymbol{z}_k)$ and $\hat{\tilde{p}}(\boldsymbol{x}_j) = \sum_k \kappa_p(\boldsymbol{x}_j - \boldsymbol{x}_k)$ for shift-invariant kernel functions $\kappa_z(\cdot)$ and $\kappa_p(\cdot)$. Then*

$$\frac{\partial}{\partial \boldsymbol{z}_i} \widehat{CS}(p(\boldsymbol{x})||q(\boldsymbol{z})) = -4 \sum_j \left[ \frac{\tilde{p}_j}{\sum_{j'} \tilde{p}_{j'}\tilde{q}_{j'}} - \frac{\tilde{q}_j}{\sum_{j'} \tilde{q}_{j'}^2} \right] \kappa_z^2(\boldsymbol{z}_j - \boldsymbol{z}_i)(\boldsymbol{z}_j - \boldsymbol{z}_i). \tag{82}$$

*Proof.* A shift-invariant kernel function satisfies $\kappa(\boldsymbol{z}_j - \boldsymbol{z}_i) = \kappa(d_{ij})$ where $d_{ij} = ||\boldsymbol{z}_j - \boldsymbol{z}_i||^2$. Hence, similar to the derivation for Proposition 2, we will have

$$\frac{\partial}{\partial \boldsymbol{z}_i} \widehat{CS}(p(\boldsymbol{x})||q(\boldsymbol{z})) = -\sum_j \left[ \frac{\tilde{p}(\boldsymbol{x}_j)}{\sum_{j'} \tilde{p}(\boldsymbol{x}_{j'})\tilde{q}(\boldsymbol{z}_{j'})} - \frac{\tilde{q}(\boldsymbol{z}_j)}{\sum_{j'} \tilde{q}^2(\boldsymbol{z}_{j'})} \right] \frac{\partial}{\partial \boldsymbol{z}_i} \tilde{q}(\boldsymbol{z}_j) \tag{83}$$

$$= -\sum_j \left[ \frac{\tilde{p}(\boldsymbol{x}_j)}{\sum_{j'} \tilde{p}(\boldsymbol{x}_{j'})\tilde{q}(\boldsymbol{z}_{j'})} - \frac{\tilde{q}(\boldsymbol{z}_j)}{\sum_{j'} \tilde{q}^2(\boldsymbol{z}_{j'})} \right] \left( \frac{\partial \tilde{q}(\boldsymbol{z}_j)}{\partial d_{ij}} + \frac{\partial \tilde{q}(\boldsymbol{z}_j)}{\partial d_{ji}} \right) (\boldsymbol{z}_j - \boldsymbol{z}_i) \tag{84}$$

$$= -\sum_j \left[ \frac{\tilde{p}(\boldsymbol{x}_j)}{\sum_{j'} \tilde{p}(\boldsymbol{x}_{j'})\tilde{q}(\boldsymbol{z}_{j'})} - \frac{\tilde{q}(\boldsymbol{z}_j)}{\sum_{j'} \tilde{q}^2(\boldsymbol{z}_{j'})} \right] 2 \frac{\partial \tilde{q}(\boldsymbol{z}_j)}{\partial d_{ij}} (\boldsymbol{z}_j - \boldsymbol{z}_i). \tag{85}$$

Note that $\frac{\partial \tilde{q}(\boldsymbol{z}_j)}{\partial d_{ij}} = \frac{\partial \kappa_z(\boldsymbol{z}_j - \boldsymbol{z}_i)}{\partial d_{ij}} = 2\kappa^2(\boldsymbol{z}_j - \boldsymbol{z}_i))$ for $\kappa_z(\boldsymbol{z}_j - \boldsymbol{z}_i)) = \left[ \frac{1}{1+||\boldsymbol{z}_i - \boldsymbol{z}_j||^2} \right]$, which is the t-distribution, or alternatively for $\kappa_z(\boldsymbol{z}_j - \boldsymbol{z}_i)) = \exp(-\kappa||\boldsymbol{z}_i - \boldsymbol{z}_j||^2)$, the Gaussian distribution. Taken together,

$$\frac{\partial}{\partial \boldsymbol{z}_i} \widehat{CS}(p(\boldsymbol{x})||q(\boldsymbol{z})) = -4 \sum_j \left[ \frac{\tilde{p}_j}{\sum_{j'} \tilde{p}_{j'}\tilde{q}_{j'}} - \frac{\tilde{q}_j}{\sum_{j'} \tilde{q}_{j'}^2} \right] \kappa_z^2(\boldsymbol{z}_j - \boldsymbol{z}_i)(\boldsymbol{z}_j - \boldsymbol{z}_i). \tag{86}$$

$\square$

**Proposition 11.** *[Cauchy-Schwarz (CS) t-SNE is a special case of MAP IT]. Let $p_{jk'}$ be the probability for the joint event $\boldsymbol{x}_j \cap \boldsymbol{x}_{k'}$. Let $q_{jk'}$ be the probability for the joint event $\boldsymbol{z}_j \cap \boldsymbol{z}_k$. If $(\boldsymbol{x}_j \cap \boldsymbol{x}_{k'}) \cap (\boldsymbol{z}_j \cap \boldsymbol{z}_k) \in \emptyset$, then*

$$CS(P_m||Q_m) = CS(P||Q). \tag{87}$$

*Proof.* We have

$$CS(P_m||Q_m) = -\log \frac{\sum_j p_j q_j}{\left( \sum_j p_j^2 \right)^{\frac{1}{2}} \left( \sum_j q_j^2 \right)^{\frac{1}{2}}} \tag{88}$$

$$= -\log \frac{\sum_j \sum_{k',k} p_{jk'} q_{jk}}{\left( \sum_j \sum_{k',k} p_{jk'} p_{jk} \right)^{\frac{1}{2}} \left( \sum_j \sum_{k',k} q_{jk'} q_{jk} \right)^{\frac{1}{2}}}. \tag{89}$$

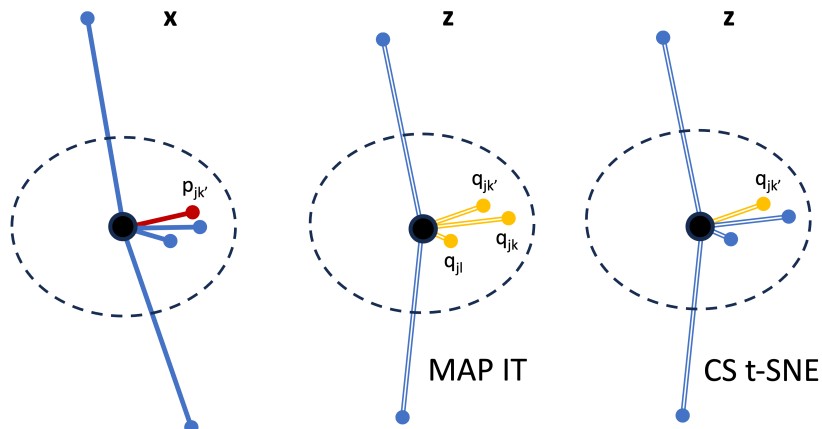

Figure 9: Illustration of CS t-SNE as a special case of MAP IT.

It suffices to look at the numerator since the terms in the denominator are related normalization quantities.

If $(\boldsymbol{x}_j \cap \boldsymbol{x}_{k'}) \cap (\boldsymbol{z}_j \cap \boldsymbol{z}_k) \in \emptyset$ then $Prob((\boldsymbol{x}_j \cap \boldsymbol{x}_{k'}) \cap (\boldsymbol{z}_j \cap \boldsymbol{z}_k)) = p_{jk'}q_{jk} = 0$ (assuming independence) for $k' \neq k$. Hence

$$\sum_j \sum_{k',k} p_{jk'}q_{jk} = \sum_{j,i} p_{ji}q_{ji} \qquad (90)$$

for $k' = k = i$. $\hfill\square$

**Comment to Proposition 11.** The illustration in Fig. 9 brings further perspective to this result.

The black filled circle denotes node $j$. Nodes within the stapled circle are assumed to be relatively near and nodes $n$ outside this circle are assumed to be distant in the sense that $p_{jn}$ is neglible for each such node $n$. For MAP IT, $p_{jk'}$ is also multiplied by probabilities $q_{jk}$ and $q_{jl}$ in addition to $q_{jk'}$ for nodes $k$ and $l$ close to $k'$. This is not the case for CS t-SNE. This shows that MAP IT is able to capture information in wider local neighborhoods compared to CS t-SNE which only captures local information via pairwise affinities, a property it shares with its KL counterpart t-SNE and methods such as Laplacian eigenmaps (Belkin & Niyogi, 2003).

In practise, this means that MAP IT models that the event $(\boldsymbol{x}_j \cap \boldsymbol{x}_{k'})$ could induce the event $(\boldsymbol{z}_j \cap \boldsymbol{z}_k)$ if $k'$ and $k$ are close.

APPENDIX C: ADDITIONAL RESULTS AND ANALYSIS

**Defining probabilities in the input space.** The projective property of the CS divergence has been shown to avoid the need for direct normalization of probabilities if the probabilities come in a form $q_j = \tilde{q}_j/Z_q$, where $Z_q$ is as explained in Section 3.2 and where one has access to $\tilde{q}_j$, and similarly for $p_j$. However, for optimization purposes, normalization $Z_p = \sum_{j'} \tilde{p}_{j'}$ is not as critical since this is a procedure done only once. In any case, it has recently been shown that dimensionality reduction methods are quite robust to the choice of input space pairwise similarity function, and even binary affinities seem to give good results for many algorithms (Böhm et al., 2022). Note that it is easy to show that even if the input space pairwise similarities do not come in the form assumed in Eq. (1), that is, $p_{ij} = \tilde{p}_{ij}/Z_p$, $Z_p = \sum_{n,m} \tilde{p}_{nm}$, but the target space similarities do, then $D_{CS}(P_m||Q_m) = D_{CS}(P_m||\tilde{Q}_m)$ (Proposition 3). In order for MAP IT to use the exact same input similarities as t-SNE commonly does, so-called symmetrized similarities are here used in the input space. These are given by $p_{ij} = \frac{p_{i|j}+p_{j|i}}{2n}$ with $p_{j|i} = \frac{\exp\left(-\kappa_i||\boldsymbol{x}_i-\boldsymbol{x}_j||^2\right)}{\sum_{n\neq i}\exp(-\kappa_i||\boldsymbol{x}_i-\boldsymbol{x}_n||^2)}$ and likewise for $p_{i|j}$. Note that the parameter $\kappa_i$ is chosen to yield a pre-specified value of the so-called perplexity of the probability distribution (see e.g. (Böhm et al., 2022)), a function of its entropy.

**MNIST.** MNIST[7] is a data set of $28 \times 28$ pixel grayscale images of handwritten digits. There are 10 digit classes (0 through 9) and a total of 70000 images. Here, 2000 images are randomly sampled. Each image is represented by a 784-dimensional vector. Figure 1 and Figure 3 in the main paper show that MAP IT produces a MNIST visualization with much better separation between classes compared to alternatives and that the embedding is robust with respect to initial conditions.

MAP IT's free parameter is the number of nearest neighbors $k$ to go into the computation of $\tilde{p}_{j_{N_{\boldsymbol{x}_i}}}$ and $\tilde{q}_{j_{N_{\boldsymbol{x}_i}}}$. Figure 10 shows representative embedding results for the subset of MNIST for different values of $k$. As in all dimensionality reduction methods, the visualization results depend on $k$. For MNIST, for $k = 7$ and for $k = 10$, the class structure appears and is relatively stable also for $k = 12$. For most data sets, a value for $k$ between 5 and 15 seem to yield reasonable results. However, the impact of this hyperparameter should be further studied in future work.

**Quantitative analysis.** For dimensionality reduction methods, various methods for quantifying neighborhood structure may provide insight. For instance, relatively low *knn recall* may illustrate better capturing of cluster structure at the potential expense of capturing manifold structures, as studied by (Böhm et al., 2022). For the embeddings shown in Figure 1, the fraction of $k$ nearest neighbors in the input space that remain among the nearest neightbors in the target space ("knn recall") is computed and shown in Table 1 for several values of $k$. For low values of $k$, PacMap and MAP IT have lower recall values, indicating that these methods better capture cluster structure and may sacrifice some ability to capture manifold structure. For $k$ ca equal to 60 and above the knn recall settles to similar values (slowly diminishing for larger $k$, not shown here).

| MNIST | | | | |
|---|---|---|---|---|
| knn | t-SNE | UMAP | PacMap | MAP IT |
| 15 | 0.48 | 0.42 | 0.37 | 0.35 |
| 30 | 0.49 | 0.47 | 0.43 | 0.41 |
| 45 | 0.50 | 0.49 | 0.47 | 0.45 |
| 60 | 0.51 | 0.50 | 0.49 | 0.48 |
| Frey | | | | |
| 15 | 0.09 | 0.08 | 0.08 | 0.09 |
| 30 | 0.11 | 0.11 | 0.1 | 0.1 |
| 45 | 0.13 | 0.13 | 0.12 | 0.13 |
| 60 | 0.15 | 0.16 | 0.15 | 0.15 |

Table 1: Illustration of knn recall.

---

[7]MNIST, Newsgroups, "Frey faces" are obtained from http://cs.nyu.edu/~roweis/data.html.

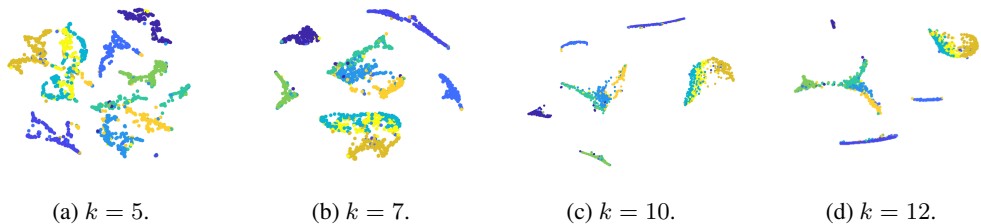

(a) $k = 5$.    (b) $k = 7$.    (c) $k = 10$.    (d) $k = 12$.

Figure 10: MAP IT for a subset of MNIST for different values of $k$.

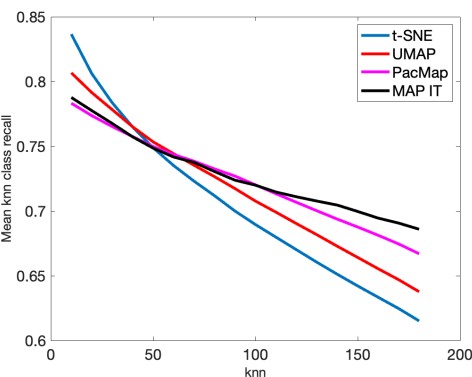

Figure 11: Class-wise knn recall.

Figure 11 shows an alternative way to quantify this effect. Here, the class-wise knn recall in the target space is shown. For each data point, the fraction of nearest neighbors in the low-dimensional space sharing class label with the point in question (Thornton, 1998) is plotted as a curve for various values of $k$. Here, the known labels for the MNIST are used. It can be observed that as $k$ increases, t-SNE and UMAP have lower class-wise knn recall compared to PacMap and MAP IT. This makes intuitive sense when compared to the visualizations provided in Figure 1.

**Further comments on MAP IT's potential for upscaling.** Figure 8 in the main paper shows the result of an initial experiment to potentially scale up MAP IT by a certain sampling procedure. The proposed sampling procedure is not the same as the one employed in LargeVis and UMAP. In those methods, attractive forces and repulsive forces are separated. The number of points to go into the computation of repulsive forces are then sampled, so-called negative sampling. When creating Figure 8 in the main paper, forces have been separated into attractive/repulsive forces resulting from nearest neighbors versus attractive/repulsive forces coming from non-neighbors. Hence, the proposed MAP IT sampling is different. Experimentally, it was observed that if the number of non-neighbor forces were downsampled for instance to $50$ percent, then a multiplication of the attractive/repulsive forces for non-neighbors by a factor two basically reproduced the original embedding. A downsampling of non-neighbor forces to $25$ percent followed by a multiplication factor of four reproduced the original embedding. Similarly, downsampling of non-neighbor forces to $12.5$ percent followed by a multiplication factor of eight reproduced the original embedding. This is illustrated in Figure 12 for MNIST with $k = 10$. In (a)-(c), the sampling is down to 50, 25, and 12.5 percent, respectively, and each of the four subfigures show the embedding after invoking a multiplication factor of $1, 2, 4$ and $8$ over the non-neighbor forces, respectively. The boxes indicate that the original embedding is in essence recreated (compare e.g. to Figure 10 (c)). For Figure 8 in the main paper, where only $3k$ non-neighbor forces are sampled, which means that ca $1.52$ percent of non-neighbors are used in the sampling, the factor used is 66 (ca $1/0.0152$). The effect that downsampling of attractive/repulsive forces for non-neighbors seems to have, requiring an inversely proportional multiplication factor as shown above, may be related to the observations made in (Böhm et al., 2022) on the effects of non-negative sampling in UMAP. Here, it was shown that negative sampling increased attraction in UMAP, and it was argued that without the negative sam-

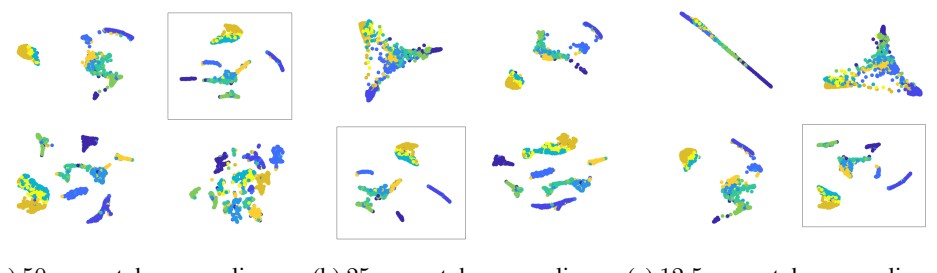

(a) 50 percent downsampling.  (b) 25 percent downsampling.  (c) 12.5 percent downsampling.

Figure 12: Each subfigure (a)-(c) show the visualization/embedding for different subsampling scenarios and for a factor 1, 2, 4, and 8, respectively, on non-neighbor attractive/repulsive forces in the MAP IT calculation.

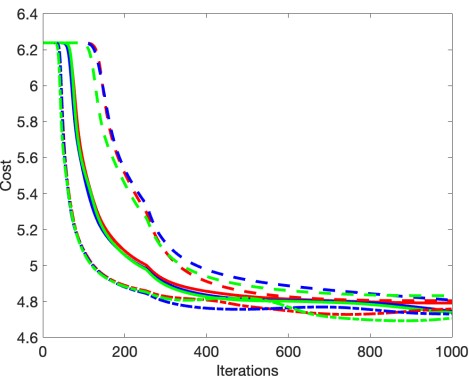

Figure 13: Illustration of MAP IT learning rates and number of iterations for a subset of USPS.

pling, UMAP may provide embeddings with less cluster structure. Obviously, for MAP IT, there are many aspects that warrant future analysis along these lines. The example provided here is meant to show that in this controlled setting, sampling of MAP IT's non-neighbor attractive/repulsive forces (Figure 8) could reproduce the original embedding (Figure 1 (d)). Please see the end of this Appendix for some further comments on implementation.

**Learning rates (USPS).** In all experiments the MAP IT learning rate has been set to 50 over 1000 iterations. Of course, changing these choices will to some degree change the embedding. These choices have however been observed to result in quite stable MAP IT results over a range of diverse data sets. Figure 13 shows the MAP IT cost as a function of iterations for different values of learning rates performed over a subset of the USPS data set (Hull, 1994). A random subset of the digits 3, 6, and 9 constitute the classes. For each learning rate $\eta$ of 25, 50 and 100 three runs of MAP IT are performed and in each case the curve of cost versus iterations is shown. For this particular data set, low cost function values are obtained quicker for $\eta = 100$ (leftmost group of curves), compared to $\eta = 50$ (middle group of curves) and $\eta = 25$ (rightmost group of curves). When approaching 1000 iterations all curves have settled at low cost function values. Further studies of the interplay between learning rate, iterations, and various design choices for the MAP IT optimization are left for future work.

**Coil 20.** This data set (Nene et al., 1996) consists of 1440 greyscale images consisting of 20 objects under 72 different rotations spanning 360 degrees. Each image is a 128x128 image which we treat as a single 16384 dimensional vector for the purposes of computing distance between images. Visualizations of Coil-20 were shown in the main paper in Figure 4. Enlarged visualizations of Coil 20 are shown in Figures 15, 16, 17 and 18.

**Visual Concepts.** Images corresponding to three different visual concepts are visualized. SIFT (Lowe, 1999) descriptors represented by a 1000-dimensional codebook for each visual concept are

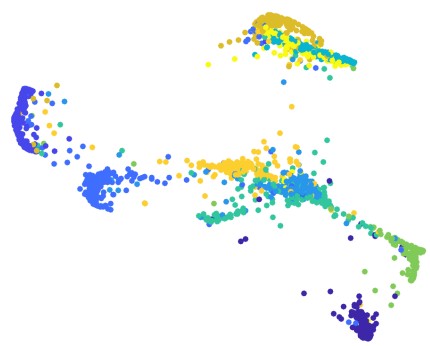

Figure 14: Preliminary experiment using the neighborhood structure in the input space when computing entropy weights for attractive and repulsive forces for MAP IT.

downloaded from the ImageNet data base (`image-net.org`) (Deng et al., 2009). The visual concepts used are *strawberry*, *lemon* and *australian terrier*. The concepts are represented by 1478, 1292 and 1079 images, respectively. The images within each category differ very much, as can be seen e.g. for *australian terrier* at `image-net.org/synset?wnid=n02096294`. A crude approach is taken here. Each image is represented by the overall frequency of codewords present for the SIFT descriptors contained in the image. Hence, each image is represented as a 1000-dimensional vector. The local modeling strength of the SIFT descriptors are lost this way, and one cannot expect the resulting data set to contain very discriminative features between the concepts. Visualizations of the visual concepts were shown in the main paper in Figure 5. Enlarged visualizations of Coil 20 are shown in Figures 19, 20, 21 and 22.

**Newsgroups.** Visualizations of words from Newsgroups were shown in the main paper in Figure 6. Enlarged visualizations of Newsgroups as word clouds are shown in Figures 23, 24, 25 and 26.

**Frey faces.** Visualizations of the Frey faces were shown in the main paper in Figure 7. Enlarged visualizations of the Frey faces are shown in Figures 27, 28, 29 and 30.

**Further comments on implementation and potential scaling.** Previously, sampling of attractive/repulsive forces associated with non-neighbors was illustrated on the recurring MNIST example and it was shown that inclusion of only a few non-neighbors in the computations can reproduce the original embedding, given the use of an appropriate multiplication factor. In the current implementation, gradients are computed directly by a matrix operation $(D - M)X$ where $M$ encodes attractive and repulsive forces according to Eq. (19) where $D$ is the diagonal degree matrix of $M$ and $X$ stores all data points as rows. Neighborhood structure is encoded into $M$. For the sampling procedure to potentially truly scale up MAP IT, more efficient knn graph implementations are needed to avoid memory issues associated with the current very basic approach. This can be done by building on efficient implementations for t-SNE and UMAP, e.g. (Böhm et al., 2022).

In the t-SNE and related literature, a problematic issue is the normalization factor $Z_q = \sum_{n,m} \tilde{q}_{nm}$ which needs to be recomputed over the optimization. For the CS divergence underlying MAP IT, the need for direct normalization is avoided. However, seemingly similar terms ($\sum_{j'} \tilde{p}_{j'}\tilde{q}_{j'}$ and $\sum_{j'} \tilde{q}_{j'}^2$) appear in the weighting of the attractive and repulsive forces, Eq. (19). The impact of these terms needs to be further studied. As a preliminary experiment, the terms are computed using only points in $N_{\boldsymbol{x}_{j'}}$-neighborhoods both in the input space and the target space for MNIST, and the embedding is shown in Figure 14. It can be seen that the main structure of the embedding is preserved.

**Further comments on parameter choices.** Note that (Belkina et al., 2019) provided advice for tailoring the early exaggeration and overall number of gradient descent iterations for t-SNE in a dataset-specific manner and argued that this may be especially important for large-scale data sets such as cytometry and transcriptomics data sets. Data sets used in this study are more traditional (MNIST etc), and available software used in this paper for t-SNE, UMAP, PacMap are already well-tested for such data sets. However, further parameter tuning as discussed in (Belkina et al., 2019)

could potentially improve results for t-SNE. Note that MAP IT uses no early exaggeration and there is much room for investigating the effect of for instance step size and the number of iterations, which may affect results also for MAP IT.

**Perspectives on MAP IT via local neighborhoods.** Section 3.4 discussed the very important role that local neighborhoods play for MAP IP, revealed via differentiation of the MAP IT cost function. The following main points were observed:

- Differentiation must be done with respect to the input space neighborhoods for all quantities. For instance, neighborhood structure in the target space $\Delta z_i$ will be given by $\Delta z_i = m(\Delta x_i)$, where $m$ is the (unknown) mapping function.
- The practical consequence of this is stated in Eq. (19), which is expressed as a sum over all points $j$. Here, the attractive force for the gradient vector $z_j - z_i$ depends on $\tilde{p}_{j_{N_{x_i}}}$ and the repulsive force depends on $\tilde{q}_{j_{N_{x_i}}}$, where $N_{x_i}$ denotes a neighborhood around $x_i$. It was further argued that for points $x_j$ that are neighbors of point $x_i$, they would already be in the neighborhood of $x_i$ such that for those nearest neighbors of $x_i$ the MAP IT update rule Eq. (19) uses $\tilde{p}_{j_{N_{x_j}}}$ for the attractive force and $\tilde{q}_{j_{N_{x_j}}}$ for the repulsive force. For points $x_j$ not neighbors of $x_i$, it was argued that $\tilde{p}_{j_{N_{x_i}}}$ simplifies to $\tilde{p}_{ij}$ for the attractive force and likewise $\tilde{q}_{j_{N_{x_i}}}$ simplifies to $\tilde{q}_{ij}$ for the repulsive force.

It is important to note that $\tilde{p}_{j_{N_{x_j}}} = \sum_{k=1}^{k_{nn}} \tilde{p}_{jk}$. It has here been emphasized that $k_{nn}$ denotes the $k$'th nearest neighbor of $x_j$ (see Section 3.2). If the $\tilde{p}_{jk}$'s are interpreted as weights on edges from node $j$ to the nodes $k = 1, \ldots, k_{nn}$, then it becomes clear that $\tilde{p}_{j_{N_{x_j}}}$ captures local properties in a region around $x_j$. All of these considerations stem from the MAP IT cost function turned into an update rule for $z_i$ via differentiation.

However, this analysis reveals a new perspective to MAP IT. If we instead of aligning the distributions over the marginal probabilities $p_j$ and $q_j$ for all $j$ by the CS divergence focus on the contribution to the marginal probabilities from the local region around point $x_j$ and $z_j$, respectively, i.e. $\tilde{p}_{j_{N_{x_j}}}$ and $\tilde{q}_{j_{N_{x_j}}}$ by

$$\underset{z_1, \ldots, z_n \in \mathbb{R}^d}{\arg\min} \ -\log \sum_j \tilde{p}_{j_{N_{x_j}}} \tilde{q}_{j_{N_{x_j}}} + \frac{1}{2} \log \sum_j \tilde{q}_{j_{N_{x_j}}}^2 \tag{91}$$

then Eq. (19) would be obtained by the same arguments as in Section 3.4 and above. This shows that MAP IT is effectively aligning the degree of a node in the input space, locally in a neighborhood, with the degree of the corresponding node locally in the target space, but where neighborhoods are defined with respect to the input space in both cases.

To some degree, this perspective resembles the local neighborhood perspective in Locally Linear Embedding (LLE) (Roweis & Saul, 2000). Also here, local neighborhoods are reconstructed in the target space versus the input space via an eigenvector operation. LLE and many other spectral methods (as mentioned in the Introduction of this paper) have inspired much of the research in visualization and neighbor embedding methods but as stated in (Wang et al., 2021) in their section *Local structure preservation methods before t-SNE* (page 6) "the field mainly moved away" from these types of methods. In the future, it would be interesting to investigate closer links and differences between the fundamental LLE idea and MAP IT.

This is an important aspect to point out for the following reason. When basing the input space similarities on symmetrized $p_{ij}$, which is done in this paper in the experimental part in order to use similarities which are comparable to what t-SNE usually process, an apparent paradox appears[8]. In that case, the form of the similarities are such that the $p_j$'s become uniform. In that case it would appear as if MAP IT is data independent, always trying to make the $q_j$'s uniform no matter what the input is. However, the above analysis highlights that via differentiation, MAP IT effectively focus on the contribution to the marginal probabilities from local regions, which will in general differ for the different nodes.

**Map KL Divergence?.** An interesting question for future research could be to investigate whether the introduction of a map KL divergence similar to the map CS divergence (Definition 4), would

---

[8]As kindly pointed out by one of the reviewers of this paper.

provide a KL-based MAP IT framework via kernel smoothing. Since the KL divergence is not projective, direct normalization of probabilities would not be avoided, but a similar-in-spirit alignment of marginal probabilities seems to be possible.

**Concluding remarks.** Together with the experiments and analysis in the main paper, these additional MAP IT results and analysis illustrate the potential of this new method to provide visualizations which in many cases are markedly different from the current state-of-the-art alternative with better class discrimination and reasonable embeddings overall, from a theoretical approach which is fundamentally different and which highlights both a viewpoint from the perspective of alignment of marginal probabilities as well as a dual viewpoint via continuous densities enabled by kernel smoothing. The role of normalization for divergences to be used as dimensionality reduction methods follows directly from the MAP IT theory.

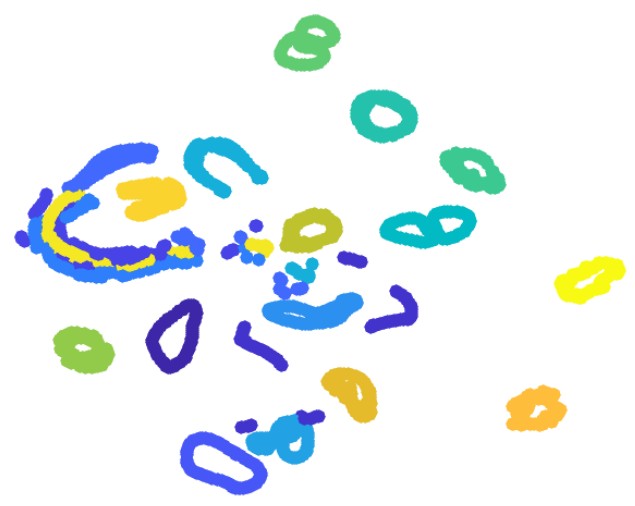

Figure 15: t-SNE embedding of Coil 20.

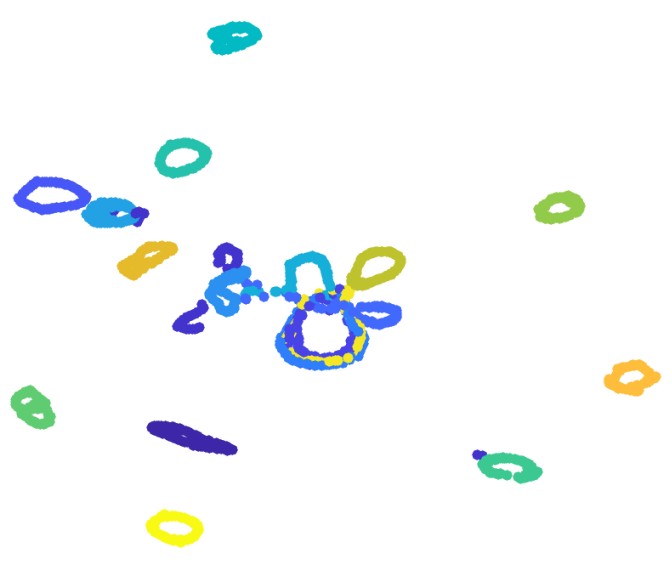

Figure 16: UMAP embedding of Coil 20.

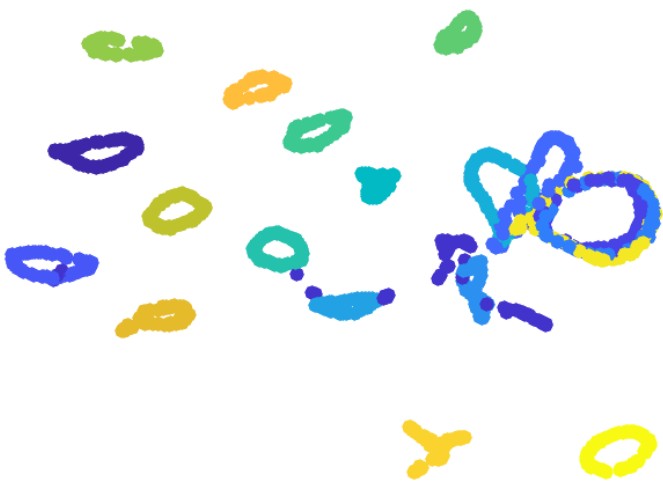

Figure 17: PacMap embedding of Coil 20.

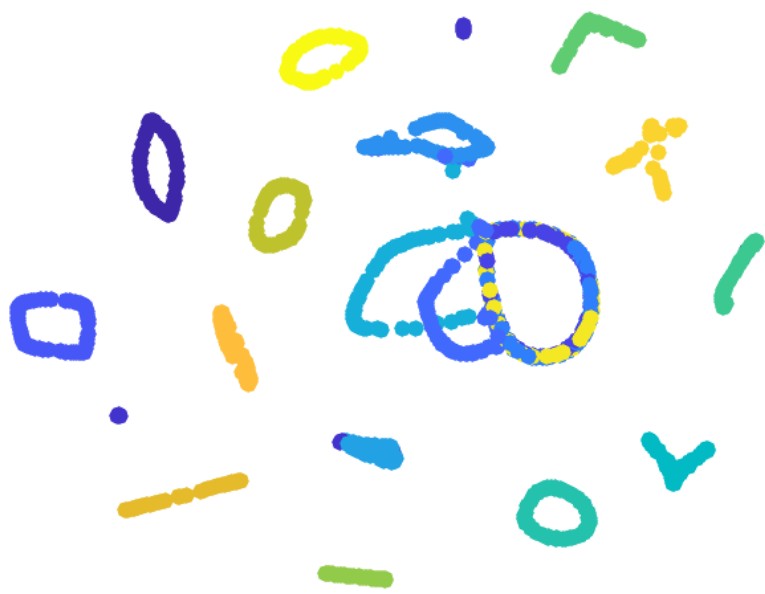

Figure 18: MAP IT embedding of Coil 20.

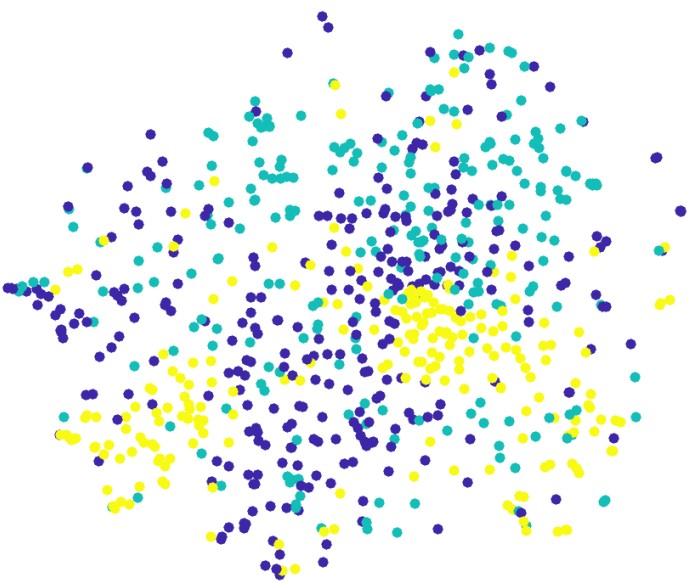

Figure 19: t-SNE embedding of visual concepts.

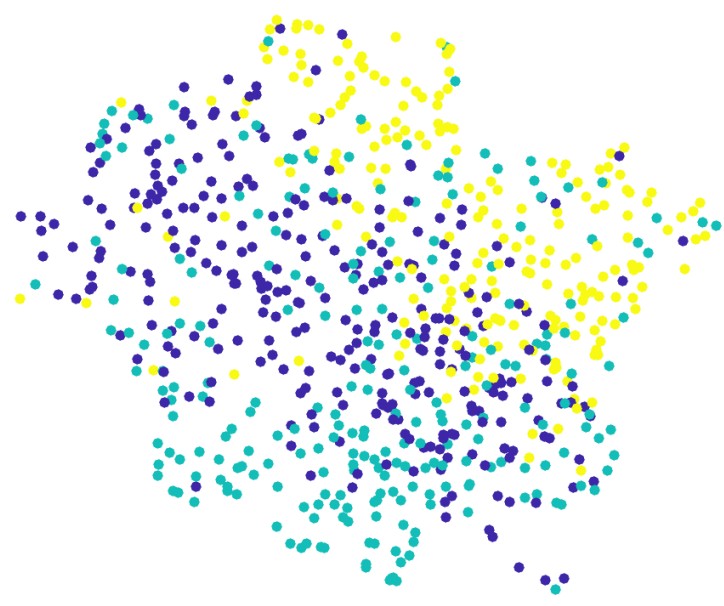

Figure 20: UMAP embedding of visual concepts..

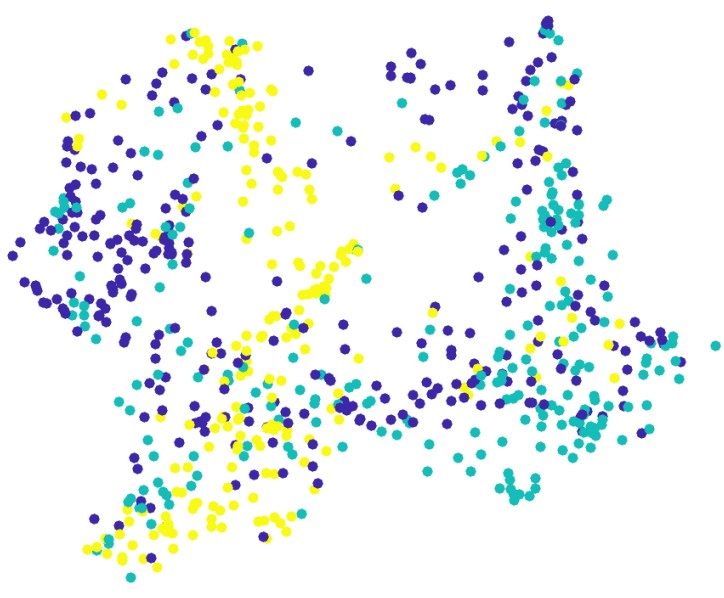

Figure 21: PacMap embedding of visual concepts..

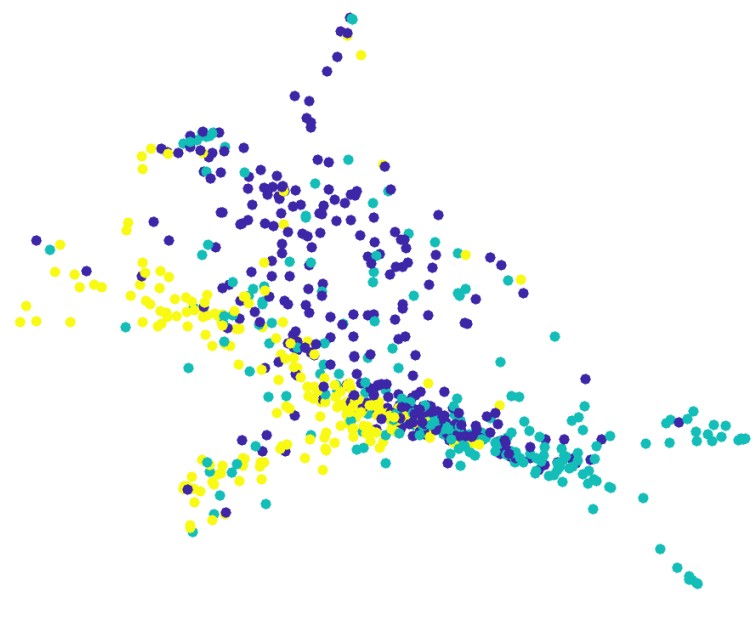

Figure 22: MAP IT embedding of visual concepts..

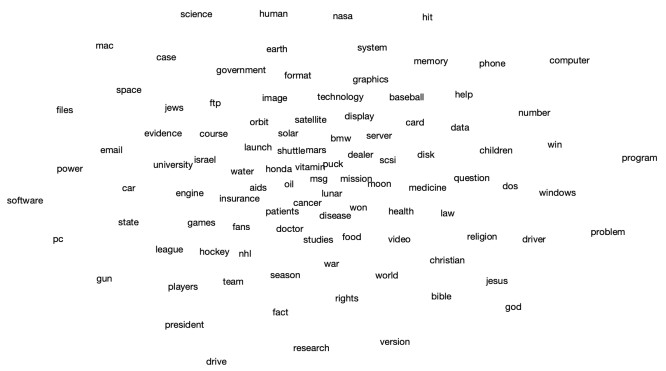

Figure 23: t-SNE embedding of words from Newsgroups.

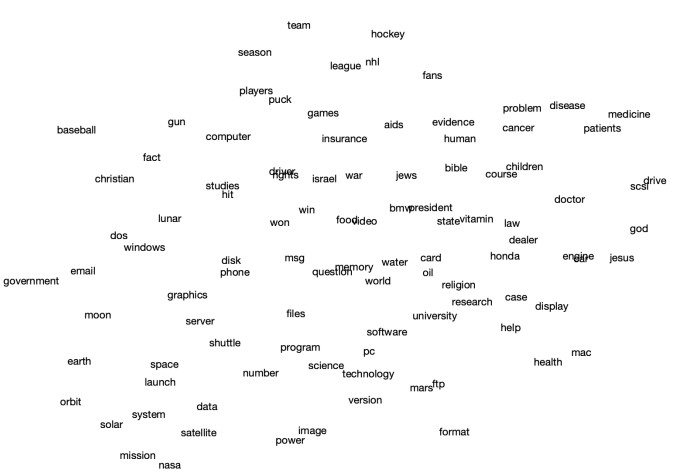

Figure 24: UMAP embedding of words from Newsgroups.

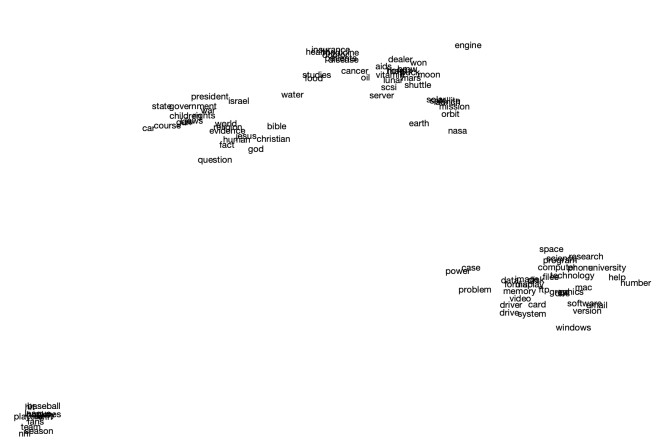

Figure 25: PacMap embedding of words from Newsgroups.

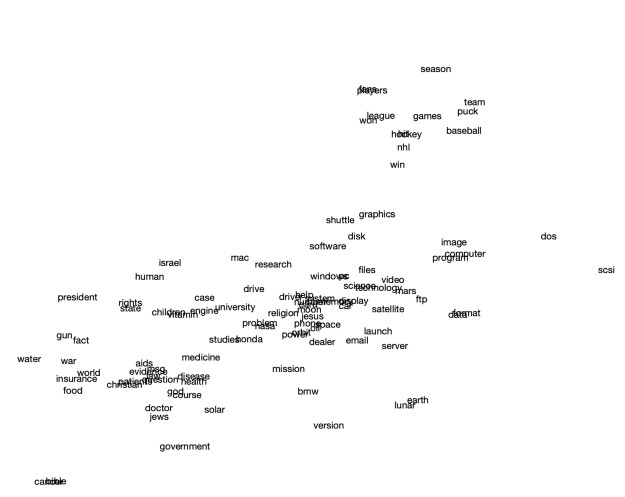

Figure 26: MAP IT embedding of words from Newsgroups..

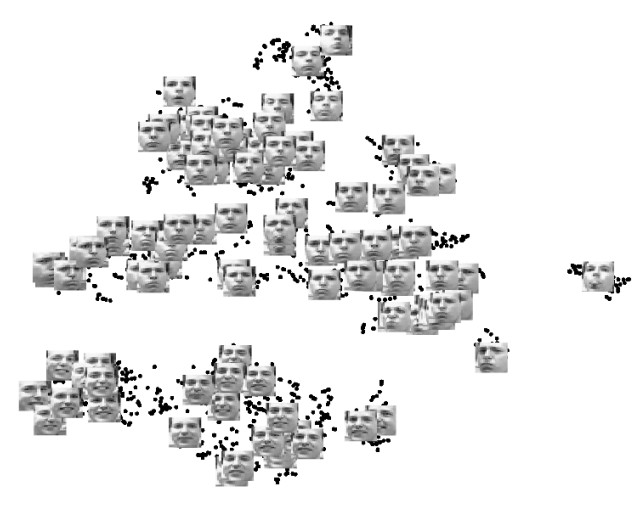

Figure 27: t-SNE embedding of Frey faces.

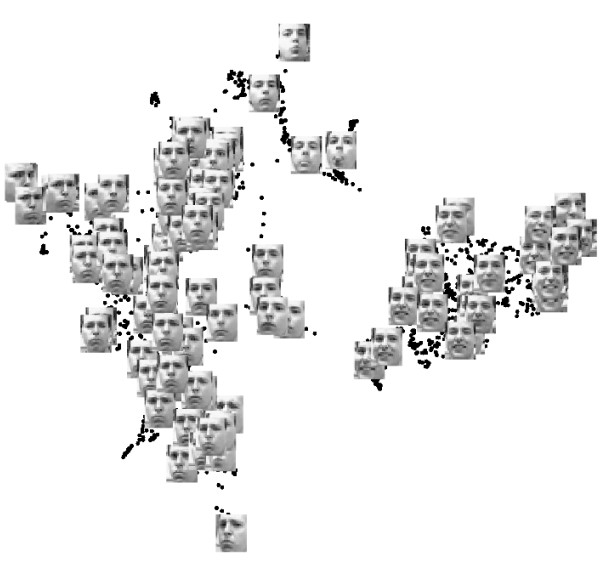

Figure 28: UMAP embedding of Frey faces.

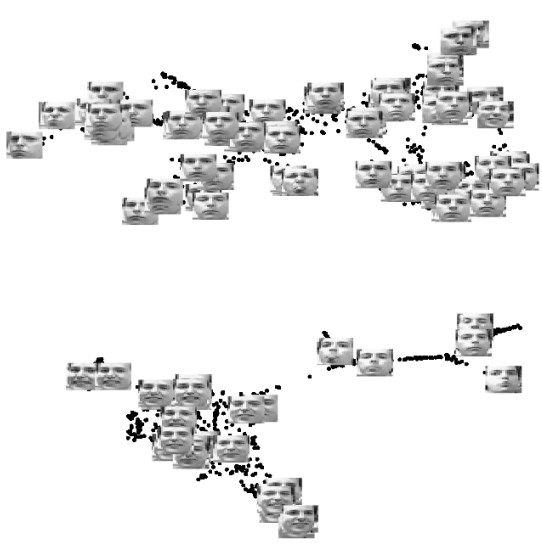

Figure 29: PacMap embedding of Frey faces.

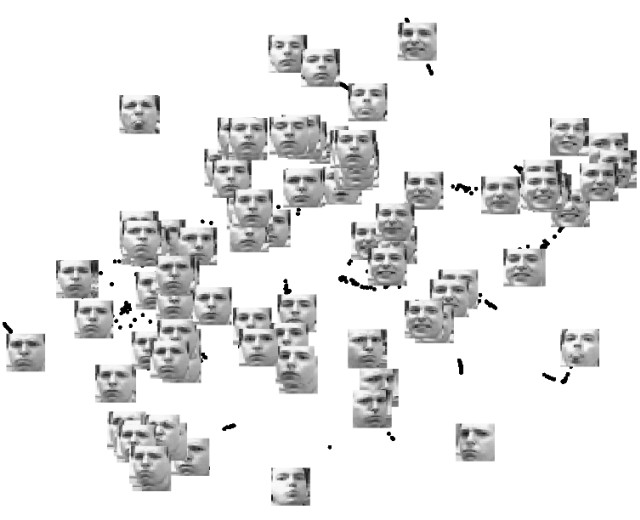

Figure 30: MAP IT embedding of Frey faces.

