# MAP IT TO VISUALIZE REPRESENTATIONS SUPPLEMENTARY MATERIAL

## 1 FURTHER PERSPECTIVES ON T-SNE, UMAP, PACMAP, AND VARIANTS

As mentioned in Section 1 (Introduction) of the main paper, a large plethora of dimensionality reduction methods exists and an excellent repository for more information is e.g. `https://jlmelville.github.io/smallvis/`. In this paper, following recent literature, the main algorithms are considered to be t-SNE (van der Maaten, 2014), and UMAP (McInnes et al., 2020), and we include the empirically motivated PacMap (Wang et al., 2021). For context, TriMap (Amid & Warmuth, 2019) and LargeVis (Tang et al., 2016) are discussed below. The t-SNE theory has been outlined in Section 2 of the main paper.

In the paper introducing UMAP, McInnes et al. (2020) argues that t-SNE should be considered the current state-of-the-art at that time, and mentions computational scalability as a main benefit of UMAP versus t-SNE. The main aspect with respect to computational scalability for UMAP versus t-SNE is that the simplical set theory shows that for UMAP normalization over pairwise similarities (probabilities in t-SNE) is not needed, as opposed to t-SNE. This illustrates the importance of the sound theoretical foundation of UMAP. As further described in (McInnes et al., 2020), UMAP's simplical set cross-entropy cost function resembles in several ways the LargeVis (Tang et al., 2016) cost function. LargeVis also avoids normalization in the embedding space, albeit from a more heuristic point of view, but not in the input space where a procedure similar to the one used in Barnes-Hut t-SNE (van der Maaten, 2014) is used. Avoiding normalization in the embedding space is key to the negative sampling strategy employed in LargeVis and which is key to its computational scalability, also an integral component in the UMAP optimization. LargeVis is not included in the experimental part of this paper to avoid clutter, but is an influential algorithm in the t-SNE family. McInnes et al. (2020) and Wang et al. (2021), for instance, have both extensive comparative experiments sections also involving LargeVis.

A motivation for Wang et al. (2021) is to discuss preservation of local structure versus global structure. They propose a heuristic method, PacMap, which is intended to strike a balance between TriMap (Amid & Warmuth, 2019) (better at preserving global structure) and t-SNE/UMAP (local structure). TriMap is is a triplet loss-based method. Wang et al. (2021) argues that TriMap is the first successful triplet constraint method (as opposed to (Hadsell et al., 2006; van der Maaten & Weinberger, 2012; Wilber et al., 2015)) but claims that without PCA initialization "TriMap's global structure is ruined". PacMap is based on a study of the principles behind attractive and repulsive forces and finds that forces should be exerted on further points and sets up a heuristically designed procedure for treating near pairs, mid-near pairs, and non-neighbors.

Understanding t-SNE versus UMAP, in particular, from a theoretical perspective, has gained interest in the recent years. Damrich & Hamprecht (2021) studies the interplay between attractive and repulsive forces in UMAP in detail and comes to the conclusion that UMAP is actually not exactly optimizing the cost function put forth in (McInnes et al., 2020). Bohm et al. (2020) studies the whole attraction-repulsion spectrum and find cases where UMAP may diverge.

Kobak & Linderman (2021) show that UMAP's initialization (Laplacian eigenmaps (Belkin & Niyogi, 2003)) is very important for UMAP's results and claims that t-SNE can be improved by a similar initialization. Wang et al. (2021) also studies initialization, and claim that both UMAP but also TriMap are very dependent on initialization. A further comment on a relationship between t-SNE and Laplacian eigenmaps is provided in Section 2 of this Supplementary material.