# OpenReview forum: "MAP IT to Visualize Representations"
_ICLR.cc/2024/Conference — ICLR 2024 poster_

### Official Review · Reviewer_L3aD · 2023-10-28

**Soundness:** 3 good
**Presentation:** 2 fair
**Contribution:** 2 fair
**Rating:** 3
**Confidence:** 5

**Summary:**

In the paper "MAP IT to visualize representations", the authors put forward a novel 2D visualisation method (MAP IT), based on t-SNE. There are two main differences between MAP IT and t-SNE: (1) MAP IT uses unnormalized affinities and replaces KL divergence with Cauchy-Schwarz (CS) divergence which is scale-invariant; (2) MAP IT applies the divergence to the "marginal" affinities obtained by summing up the rows of the pairwise affinity matrix. The authors claim that this results in a visualization method which is conceptually different from all t-SNE-like predecessors. The authors use several small datasets (n <= 2000) including a subset of MNIST to argue that MAP IT outperforms t-SNE and UMAP.

**Strengths:**

I initially found the paper interesting, as it shows a good understanding of existing 2D visualization methods (t-SNE/UMAP) and develops what seems to be a novel alternative approach. I also agree with the authors that their MAP IT visualization of n=2000 subset of MNIST looks interesting and very different from t-SNE/UMAP.

**Weaknesses:**

Unfortunately, in the end I was confused by the presentation and not convinced by the paper. One big issue is that the paper is hard to understand: formulas have typos, many terms are not properly defined, overall section structure is sometimes confusing, etc. Another big issue is that the authors only show results on tiny datasets and do not seem to have a working scalable implementation that would allow embedding even a modest-sized full MNIST, let alone datasets with millions of points. The third issue is complete lack of quantitative evaluations.

**Questions:**

MAJOR ISSUES

* section 3.1: Z_p and Z_q are never formally defined. While I understand that Z_q is the sum over all N^2 Cauchy kernels, I am not sure what Z_p is in t-SNE. Is it simply 2n?

* MOST IMPORTANT QUESTION: The authors define marginal \tilde p_j as sum over rows of the unnormalized t-SNE affinity matrix. They don't actually define \tilde p_ij, but I assume that it's just p_ij * 2n. Note that in t-SNE p_{i|j} values sum to exactly 1 in each row, by construction. p_{ij} are obtained by symmetrizing p_{i|j}, but approximately they still have constant row and column sums. So when the authors define marginal p_j probabilities, to me it seems they are constant (??!) or at least near-constant. So I don't undestand how this can be useful for any dimensionality reduction algorithm. What am I missing?

* Conceptual question 1. The authors put a lot of emphasis on using unnormalized affinities (and replacing KL with CS) and also on using marginal probabilities instead of pairwise probabilties. Are these two things somehow related? Could one use unnormalized affinities and CS loss with the pairwise affinities? Could one use marginal probabilities in the KL loss? Are these two independent suggestions and MAP IT just happens to implement both, or are these two suggestions somehow related and follow from each other?

* Conceptual question 2. The authors put a lot of emphasis on using unnormalized affinities, but their CS loss function performs normalization within the loss function (Equation 5). This normalization leads to the N^2 repulsion term similar to t-SNE (Equation 8). To me this seems like the authors simply "moved" the normalization from one place (affinities) to another place (loss function), but nothing much changed compared to t-SNE. What am I missing?

* The beginning of section 4 says that MAP IT uses perplexity 15. But in caption of Figure 3 and later in the text (esp. Appendix C), the authors mention k=10 value as if MAP IT uses kNN graph with k=10. How does perplexity 15 relate to k=10? This is very confusing.

* Major limitation: the authors only have implementation that allows them to embed n=2000 data sets. That was fine 25 years ago, but in 2023 this is a major limitation. What is confusing to me, is that in Figure 8 the authors show that they can reduce the runtime 100x fold, and still obtain nearly the same result, but they only show it on the same n=2000 subset of MNIST. Why not run this 100x-sped-up approximation on the entire MNIST? That would be interesting to see!

* Major limitation: no quantitative evaluation. E.g. for MNIST one could do kNN classification in the 2D space, and compare MAP IT with t-SNE/UMAP. One could also compute various NN preservation metrics. Would of course be much more interesting to do this on the entire MNIST and not on the n=2000 subset...


MINOR ISSUES

* page 2, formula (1): the minus sign is missing after the last equality sign

* page 2, formula for p_ij in t-SNE: there should be 2n in the denominator, not just 2.

* page 3, "the role of normalization": see Damrich et al 2023 https://openreview.net/forum?id=B8a1FcY0vi for parametric t-SNE. This paper would be important to cite in various places, including in the Appendix A

* page 3: "marginal probabilities" appear in Definition 2 but have not been properly defined yet.

* section 3.2, second line: one of the p_j = should be q_j =.

* section 3.3: this section needs some introduction. By the end of section 3.2 it seems that the method is already defined. So why do you need another treatment in section 3.3? This needs more motivation.

* page 14: relationship between t-SNE and Lapl. Eigenmaps was discussed in https://jmlr.org/papers/v23/21-0055.html and https://epubs.siam.org/doi/10.1137/18M1216134, it seems one should cite them here. And "not to have been discussed much in the literature" is not exactly right.

---

> ### Author Response · Authors · 2023-11-20
> **Quantitative evaluation added, normalization defined, questions addressed**
>
> Dear reviewer L3aD,
>
> Thank you for your review! I have marked changes in **blue** in the revised manuscript.
>
> **Section 3.1, $Z_q$ and $Z_p$ not formally defined**: In order to motivate the theory in Section 3 on CS as a projective divergence, I take as a starting point $q_{ij} = \left[ 1+||\vec z_i-\vec z_j||^2 \right]^{-1} / Z_q$ where $Z_q = \sum_{n, m} \left[ 1+||\vec z_n-\vec z_m||^2 \right]^{-1}$ is an explicit normalization factor.  Furthermore, $p_{ij} = \exp\left(-\kappa_i ||\vec x_i-\vec x_j||^2 \right)/ Z_p$, with normalization factor $Z_p = \sum_{n, m} \exp(-\kappa_i ||\vec x_n-\vec x_m||^2)$. This is now made explicit on **page 2**.
>
> **Most important question**: With the above starting point and definitions, the MAP IT cost function will basically align the degrees of corresponding nodes in the target space versus the input space. However, in order to use the same way to create input space similarities for t-SNE and MAP IT, to have a similar underlying foundation, I discuss in **Appendix C, page 22** that in practise symmetrized probabilities are used such that $p_{ij} = \frac{p_{i|j} + p_{j | i}}{2n}$ with $p_{j|i} = \frac{\exp\left(-\kappa_i ||\vec x_i-\vec x_j||^2  \right)}{\sum_{n \neq i} \exp(-\kappa_i ||\vec x_i-\vec x_n||^2)}$ and likewise for $p_{i|j}$. The projective property with respect to $Z_q$ is still in force. You are right that this basically creates unit node degrees in the input space and thus more uniform marginal input space probabilities. This means that the target space marginal probabilites will be optimized towards a more uniform distribution too. The node degree is a quantity that relates a point to all other points it is connected by the strength (weights) on the connections. A node degree forced to be one seems to me as a means to avoid very small weights or very strong hubs and such properties should be beneficial to transfer to the target space.
>
>  **Lack og quantitative evaluations**: I agree that this should have been included from the start, thanks. I included both results for knn recall (input space vs. target space) and for class-wise knn recall in the target space. You can find this on **Appendix C, page 22** in the revised manuscript.
>
>  **Conceptual question 1**: Thanks for raising these conceptual questions. The use of a projective divergence is what makes direct normalization of probabilities unnecessary, either for a cost function only based on pairwise affinities or for a cost function based on marginal affinities. Actually, in Proposition 11 I do show that MAP IT as a special case reduces to the CS divergence used with only pairwise affinities. However, your question regarding potentially having marginal probabilities for the KL loss is intriguing. In that case, the direct normalization of probabilities would be needed since the KL is not projective. I added a paragraph in **Appendix C, page 26** where I indicate that in future work, Definition 4 may be extended to a *map KL divergence*.
>
> **Conceptual question 2**: Thanks for pointing out the N^2 term. As mentioned to reviewer vQ36, the use of the CS divergence to derive MAP IT does render explicit normalization of probabilities unnecessary. But as you point out, a quadratic complexity still arises in the (entropy) weighting  and in future work the impact of these weighting factors and how they relate to each other should be further studied. I toned down the claim about MAP IT being inherently scalable in Abstract, etc, for that reason. I did perform a preliminary analysis where I computed these quantities by only using nearest neighbors of points in the computation, where the nearest neighbors are defined with respect to the input space. This is explained on **page 25** and the result is shown in **Figure 14**. This relates to the discussion in Section 3.4 **MAP IT via local neighborhoods**.
>
> **Perplexity**: The perplexity is related to the Gaussian bandwith $\kappa_i$ parameter for computing $p_{ij}$. It is not related per se to the use of $k=10$ nearest neighbors for the computation of the attractive and repulsive forces, as discussed in Figure 3 and in Appendix C. However, I included in **Appendix C, page 22** a more detailed discussion of the definition of perplexity and the $\kappa_i$.
>
> **Scalable implementation**: In **Appendix C, page 25** implementation details are discussed. The current code is not very efficient, and I can of course only agree that it would have been better to have a large-scale implementation. Hopefully, there is room to have the fundamentally different ideas that MAP IT represents and the current demonstrations published even if the implementation can still be improved.
>
> **MINOR issues** Thanks for these good suggestions! They have all been implemented/fixed. For instance, including a motivation in Section 3.3, discussing Damrich (2023), discussing t-SNE vs Laplacian eigenmaps in **Appendix A**.

---

> ### Comment · Reviewer_L3aD · 2023-11-20
>
> Thank you for your response. I am only going to respond to my main confusion, because I am afraid I still don't follow:
>
> > With the above starting point and definitions, the MAP IT cost function will basically align the degrees of corresponding nodes in the target space versus the input space.
>
> Just to clarify. The way you define p_ij on page 2 ensures that marginal p_i (row sums) will all be exactly identical (because kappa_i values are adjusted to make each row sum to 1). Is that right? (I understand that in practice you use symmetric affinities which are slightly different, but as you say this is not important.)
>
> If so, your loss function is DATA INDEPENDENT: q_i's need to match a constant vector of identical values p_i, whatever the original dataset was!
>
> I don't understand how this can possibly result in anything useful. In fact embeddings of all datasets with the same sample size should be all identical. But clearly this is not the case in the figures in your paper. How come?
>
> > You are right that this basically creates unit node degrees in the input space and thus more uniform marginal input space probabilities. This means that the target space marginal probabilities will be optimized towards a more uniform distribution too. The node degree is a quantity that relates a point to all other points it is connected by the strength (weights) on the connections. A node degree forced to be one seems to me as a means to avoid very small weights or very strong hubs and such properties should be beneficial to transfer to the target space.
>
> Here you are arguing that the fact that "target space marginal probabilities will be optimized towards a more uniform distribution" is a good thing and not a bad thing. But there is no other term in the loss function! That's all there is that you are optimizing. If the entire algorithm is: "make q_i values uniform", then how can it work?

---

> > ### Author Response · Authors · 2023-11-22
> > **Explanation for the apparent paradox on uniform marginal probabilities**
> >
> > Dear L3aD,
> >
> > **Thanks** for taking the time to reiterate your concern. This is much appreciated, since I probably didn't fully grasp all aspects of it at first. I agree about your concern that MAP IT would appear to be data independent for symmetrized similarities.
> >
> > I have provided an analysis, highlighted in **magenta** with the heading **Perspectives on MAP IT via local neighborhoods** on **page 26** in the revised manuscript. It spans 3/4 of that page. The analysis end with
> >
> > "... This shows that MAP IT is effectively aligning the degree of a node in the input space, locally in a neighborhood, with the degree of the corresponding node locally in the target space, but where neighborhoods are defined with respect to the input space in both cases.
> >
> > This is an important aspect to point out for the following reason. When basing the input space similarities on symmetrized $p_{ij}$, which is done in this paper in the experimental part in order to use similarities which are comparable to what t-SNE usually process, an apparent paradox appears$^\dagger$. In that case, the form of the similarities are such that the $p_j$'s become uniform. In that case it would appear as if MAP IT is data independent, always trying to make the $q_j$'s uniform no matter what the input is. However, the above analysis highlights that via differentiation, MAP IT effectively focus on the contribution to the marginal probabilities from local regions, which will in general differ for the different nodes."
> >
> > $^\dagger$ As kindly pointed out by one of the reviewers of this paper.
> >
> > I hope this sheds light on your concern, and I thank you once again.

---

> > > ### Comment · Reviewer_L3aD · 2023-11-22
> > >
> > > Thank you for you reply and for further edits to the paper.
> > >
> > > Unfortunately I have to say that I am not convinced by this explanation. I don't really understand it. As formulated, your algorithm simply should not work (the loss function strives to make q_i uniform, which is useless). Obviously it does work, but it is entirely unclear why. Moreover, the algorithm does not scale beyond O(1000) points, and the numerical improvement in quantifiable benchmarks is absent (kNN recall) or not very impressive ("class-wise kNN recall" but only for very large $k$ values).
> > >
> > > All in all, I cannot recommend this paper for acceptance in its current form, and unfortunately have to keep my negative score (3).

---

> > > > ### Author Response · Authors · 2023-11-23
> > > > **Respectfully disagree on "should not work"**
> > > >
> > > > Dear L3aD,
> > > >
> > > > I thank you for your comments. However, I respectfully disagree with you on "should not work". The loss function in general does not strive to make q_i uniform, but in the special case of using symmetrized affinities (to try to make a close comparison to t-SNE) the apparent paradox arises. But as explained in Appendix C in a detailed analysis, even then (and strongly connected to Section 3.4 "MAP IT via local neighborhoods") it becomes clear that by derivation, attractive forces and repulsive forces via p_j and q_j are always related to local neighborhoods, and the effective cost function even then never tries to aim for q_j to be uniform, even in this case of symmetrized affinities.

---

> > > > > ### Comment · Reviewer_L3aD · 2023-11-23
> > > > > **Remaining concerns**
> > > > >
> > > > > > in the special case of using symmetrized affinities (to try to make a close comparison to t-SNE) the apparent paradox arises.
> > > > >
> > > > > But that is what you use for all visualizations (Figures 1/3/5/6/7/8), right? So it's not really a "special case" but the only case that is showcased in your paper.
> > > > >
> > > > > > But as explained in Appendix C [...] the effective cost function even then never tries to aim for q_j to be uniform
> > > > >
> > > > > As I wrote, I did not really understand this explanation unfortunately, and more generally I don't understand why the "effective cost function" is different from the "stated cost function". You talk about gradients etc., but this does not help me understand why the effective cost arises at all. To me, "effective" would mean that the algorithm is NOT optimizing the same loss function as stated. Is this the case?
> > > > >
> > > > > More generally, give that basically ALL figures in the paper are based on this algorithm, this discussion should not be in Appendix C, but play a central part in the paper, IMHO.

---

### Official Review · Reviewer_vQ36 · 2023-10-31

**Soundness:** 2 fair
**Presentation:** 2 fair
**Contribution:** 2 fair
**Rating:** 5
**Confidence:** 4

**Summary:**

The authors introduce a new visualization method, called MAP IT.  The
method supposedly improves over other visualization methods that are
commonly used, such as t-SNE, UMAP, etc.

**Strengths:**

The visualizations show some features that are better highlighted
compared to other methods.

The method seems straightforward to optimize and does not rely on too
many hyperparameters.

**Weaknesses:**

The method does not seem scalable, contrary to the authors claim.  The
CS divergence includes a summation over all p_j and q_j, hence you
have quadratic complexity when optimizing.

Continuing the point, there are no large-scale visualizations.  While
the authors claim that their approach could be scaled, I am sceptical
of it previsely because of the definition of the CS divergence.

Misc:

some citations could be added, e.g. on p. 3 (The role of
normalization) you could cite NCVis (Artemenkov & Panov, 2021,
https://arxiv.org/abs/2001.11411); an approach that estimates the
norm. const. via NCE (Gutmann & Hyvarinen, 2012).  The role of the
normalization constant is also discussed in Böhm et al. (2022, JMLR)
as well as Damrich et al. (2023, ICLR).

Talking about the scalability of the method is fine, but there is no
large-scale visualization that demonstrates that the method scales
beyond what other methods can esily visualize today.

In the same vein, there are no timings reported for any of the
methods.

**Questions:**

Could you comment on the computational complexity?

Why did you not chosse to use commonly used optimization parameters
for t-SNE?  In Belkina et al. 2019 they highlight what approaches work
well and they considerably improve the visualization quality.  This
would also improve the comparison that you draw from Figure 1.

Why did you choose the delta-bar-delta method for optimization?

Why do you think current approaches are not sufficient for the
datasets that you show in the paper?

---

> ### Author Response · Authors · 2023-11-19
> **Comment on quadratic term, optimization parameters**
>
> Dear reviewer vQ36,
>
> Many thanks for your review! Your review was very useful to me, also pointing out some literature which I hadn't cited (Artemenkov, Panov) and also making me go deeper into papers I had already cited (Böhm et al. (2022) and Damrich et al. (2023).
>
> **Quadratic complexity**: The use of the CS divergence to derive MAP IT does render explicit normalization of probabilities unnecessary. But as you point out, a quadratic complexity still arises in the (entropy) weighting of the attractive and repulsive forces and in future work the impact of these weighting factors and how they relate to each other should be further studied. I toned down the claim about MAP IT being inherently scalable in Abstract, etc, for that reason. I did perform a preliminary analysis where I computed these quantities by only using nearest neighbors of points in the computation, where the nearest neighbors are defined with respect to the input space. This is explained on **page 25** and the result is shown in **Figure 14**. This relates to the discussion in Section 3.4 **MAP IT via local neighborhoods**. This is however a question which merits further work and analysis.
>
> **Timings**: On **page 25** it is explained that the current implementation is based on a matrix operation $(\boldsymbol{D}- \boldsymbol{M}) \boldsymbol{X}$  where $\boldsymbol{M}$ encodes attractive and repulsive forces according to and where $\boldsymbol{D}$ is the diagonal degree matrix of $\boldsymbol{M}$. The matrix $\boldsymbol{X}$ stores all data points as rows. I am working on a more efficient knn graph implementation, although the main idea and concept is conveyed. This is why timing is not explicitly reported.
>
> **Suggested papers**: I have included the suggested papers both in **Section 2, page 3**, in **Appendix A, page 13** and other places too.
>
> **Optimization parameters for SNE, Belkina et al. (2019)**: Thanks for pointing me to this paper. I appreciate it a lot. Since the benchmark datasets studied in the paper are quite typical of the type of datasets that state-of-the-art implementations of t-SNE, UMAP, PacMap, etc, have been developed for an optimized for with their hyperparameters, I opted for those. I have however on **page 25** included a paragraph where I discuss Belkina et al. (2019). I agree that it may be that e.g. t-SNE could perhaps benefit from some more parameter tuning. At the same time, I tried to keep MAP IT very clean, not even using early exaggeration to not end up not optimizing a true divergence, and to stay close to the basic t-SNE type of approach. MAP IT could probably be improved by optimizing hyperparameters more.
>
> **Delta-bar-delta**: I chose the delta-bar-delta since it was common in the t-SNE literature. I observed slightly faster growing of structure in the MAP IT embedding when using this approach.
>
> **Current approaches not sufficient?**: Thank you for this question, I can see why you would ask it. I am not necessarily saying that the current approaches are not sufficient. But I observe some differences for MAP IT compared to the alternatives, and I think it is because MAP IT is fundamentally different in its approach.
>
> Thanks again, I appreciate your review and your efforts!

---

> > ### Comment · Reviewer_vQ36 · 2023-11-23
> >
> > Thank you for the response.  I appreciate that the paper now includes further experiments, but I am afraid that the current state is not sufficient to recommend it for acceptance.  Looking at the additional visualizations in the appendix it is not clear in all of the datasets that MAP IT performs better than contemporary approaches.
> >
> > The knn recall experiment feels a bit weak since the recall will obviously improve and in fact it will converge to 1 when k tends to the dataset size.  On the flipside, you could report some metrics for preservation of global structure such as the correlation of distances between data points in the target and output space.  That would be one way to quantify global structure.
> >
> > As it stands, I think the paper needs to be reworked/restructured substantially to add quantitative evidence that the method performs better than already established approaches, as has also been highlighted by reviewer BmiL.
> >
> > While the code does not have to be fully developed before a submission, I think there are quite a few optimization techniques that should be applicable to bring down the quadratic scaling of the algorithm to something more acceptable, such as O(n log n).

---

### Official Review · Reviewer_LbLf · 2023-11-01

**Soundness:** 3 good
**Presentation:** 3 good
**Contribution:** 3 good
**Rating:** 8
**Confidence:** 4

**Summary:**

This paper introduces an approach to dimensionality reduction by Cauchy-Schwarz projective divergence called MAP IT. This methodology aligns discrete marginal probability distributions rather than individual data points.

**Strengths:**

* This paper presents a new perspective on dimension reduction using projective divergence.
* The manuscript is well written, with most equations explained clearly and easy to follow.
* The authors demonstrate their method across various data sets, showing its ability to discover high-dimensional structures and visualize them in a lower dimension.
* The commitment to release the code publicly after the review process is admirable.

**Weaknesses:**

The paper presents projective divergence as a method to simplify high-dimensional data, with its main advantage being the removal of weight normalization. However, from the authors experiments, it's unclear how this could be an improvement compared to existing methods.

**Questions:**

There are some typos in the manuscript:
1. Sixth paragraph: normaliztion
1. After equation 18: neigborhood
1. After equation 19 second line: also i the
1. After figure 3: neightbors
$D(P||Q) = D(P˜||Q˜), \forall Z_p,Z_q\neq0$, could be better to understand over the mention of "$Z_p$ and $Z_q$ being normalizing constants" which implies $Z_p$ and $Z_q$ need to be some specific values.

---

> ### Author Response · Authors · 2023-11-19
> **Fixed typos, added explanation of normalization constants**
>
> Dear reviewer LbLf,
>
> Let me first thank you for the review and for appreciating my efforts to provide a new perspective to dimension reduction. This is much appreciated. I have made several changes in the manuscript and in the experimental part to try to make the paper better. I have marked changes in **blue** in the revised manuscript.
>
> **Typos**: I thank you for pointing out the typos. I have fixed these and I hope that I have avoided adding new ones as I have worked on the revision.
>
> **Normalization constants**: I realized when reading your comment, and also when reading comments by reviewer L3aD, that I should have been more precise. I have now added definitions for the normalizing constants $Z_q$ and $Z_p$ in relation to **Eq. (1), page 2**, in **Definition 1**, just before **Eq. (7)** as well as several places in **Appendix C**.
>
> Thank you again for your review!

---

### Official Review · Reviewer_BmiL · 2023-11-02

**Soundness:** 3 good
**Presentation:** 3 good
**Contribution:** 3 good
**Rating:** 8
**Confidence:** 2

**Summary:**

The authors propose MAP IT, a new algorithm for manifold learning / dimensionality reduction for visualization. They compare qualitatively against previous approaches, including UMAP and T-SNE.
Their approach follows t-SNE but replaces the KL divergence with the Cauchy Schwarz Divergence, which removes the need for normalization, resulting in an approach that is theoretically more scalable.

FWIW, I did not have the time to check the proofs, and I don't think I am familiar enough with the literature on relating t-SNE and UMAP to properly appreciate the details in the paper.

**Strengths:**

The paper provides a new algorithm for a common task, visualizing high dimensional datasets by projecting to two dimensions. They provide an in-depth discussion of how their approach relates to previous approaches, in terms of core ideas, computation, and results. The appendix contains helpful discussion and additional experiments, and all experiments are outlined in great detail.

**Weaknesses:**

I have two main critiques of the paper:
- It does not provide quantitative results. It's common in this area to report 1NN results over various datasets in the projected space. While this is a somewhat arbitrary metric, it seems to be the best there is so far, and I think it would be prudent to include it.
- The paper claims two conceptual novelties, the consideration of neighborhoods and the lack of normalization. Neither of these seem to me as new as the paper claims. Normalization is also not required in UMAP (as the paper mentions). Considering local neighborhoods is done explicitly in the less recent LLE, and implicitly in spectral embedding and laplacian eigenmaps. I was under the impression that t-SNE also has a similar "dual" interpretation in terms of kernel density, but I might be mistaken.

Minor:
At the bottom of page 1, "normalization" is misspelled "normaliztion".

**Questions:**

Can you explain why the kernel smoothing view doesn't apply to t-SNE?

---

> ### Author Response · Authors · 2023-11-19
> **Added quantitative results and commented on local neighborhoods and kernel smoothing**
>
> Dear reviewer BmiL,
>
> I sincerely thank you for the review. This is much appreciated. I have made several changes in the manuscript and in the experimental part in order to accommodate your suggestions and comments. I have marked changes in **blue** in the revised manuscript.
>
> **Quantitative results**: Thank you very much for proposing to add quantitative results. I agree that this should have been included from the start. I included both results for knn recall (input space vs. target space) and for class-wise knn recall in the target space. You can find this on **page 22** in the revised manuscript. The quantitative results for MNIST shed light on MAP IT showing that the method is probably somewhat better at capturing cluster structure compared t-SNE and UMAP, maybe at the expense of capturing more continuous manifold-like structures. This makes intuitive sense when looking at Figure 1.
>
> **Normalization**: You are right that normalization is not required by e.g. UMAP and I tried to be a bit more specific to say that the MAP IT theory shows when normalization is not needed when a statistical divergence is the base for the dimensionality reduction. For instance, in Appendix C, concluding remark, it now reads *The role of normalization for divergences to be used as dimensionality reduction methods follows directly from the MAP IT theory.*
>
> **Local neighborhoods**: Preservation of local neighborhoods is also a key part of e.g Laplacian eigenmaps and t-SNE itself, but via pairwise similarities only. For instance, on **page 21** I included the sentence *this shows that MAP IT is able to capture information in wider local neighborhoods compared to CS t-SNE which only captures local information via pairwise affinities, a property it shares with its KL counterpart t-SNE and methods such as Laplacian eigenmaps \citep{Belkin2003}.*
>
> **t-SNE and kernel density**: Actually, in Bunte, Haase and Villmann (2012) "Stochastic neighbor embedding (SNE) for dimension reduction and visualization using arbitrary divergences" a form of t-SNE kernel density variant was discussed, but this it quite different from MAP IT.
>
> However, I have been pondering your question "Can you explain why kernel smoothing doesn't apply to t-SNE?" I think that similar to the new quantity I propose, which I called the map CS divergence (Definition 4) a map KL divergence can also be possible. This would probably yield a related KL-based interpretation, but it would be in terms of the marginal probabilities and not the pairwise similarities which standard t-SNE optimizes. I added a paragraph about this on **page 26**, as a potential interesting avenue for future research.
>
> **Thanks again!**

---

> > ### Comment · Reviewer_BmiL · 2023-11-22
> > **Quantitative results and neighborhoods**
> >
> > Thank you for the response and the additional experiments. Doing new experiments as part of a revision phase is definitely a major effort that I appreciate. However, I don't think the quantitative experiments that you provide are on par with experiments in other related papers; it would be great to have several datasets, not just one.
> > Also, regarding your claim that MapIT captures manifold structure as opposed to cluster structure, it would be great to substantiate the claim in a more quantitative, or at least less subjective way.
> >
> > Finally, I agree that Laplacian Eigenmaps and t-SNE only capture local neighborhoods implicitly via pairwise relationships, but LLE is in fact more explicit about the modeling of neighborhoods than MAP-IT. While I haven't spent enough time with your analysis to have a strong opinion on the difference between t-SNE and MAP-IT when it comes to modeling neighborhoods, I think the claim that modeling whole neighborhoods is a novel perspective in manifold learning is overstated, given the existence of LLE.

---

> > > ### Author Response · Authors · 2023-11-23
> > > **MAP IT captures cluster structure, developed for visualization + comment on LLE**
> > >
> > > Dear BmiL,
> > >
> > > Thank you for your comment.
> > >
> > > I do agree that LLE is an important method and it was cited in the paper from the very beginning and mentioned explicitly in footnote 3: "t-SNE and the other methods mentioned here all use gradient descent but build on many aspects from spectral theory, e.g. (Roweis & Saul, 2000), (Tenenbaum et al., 2000), (Belkin & Niyogi, 2003), (Donoho & Grimes, 2003), (Jenssen, 2010)." LLE has been shown in several papers to not yield visualizations of the same quality as t-SNE, UMAP, PacMap.
> > >
> > > But I agree with you that LLE should be mentioned in a more concrete manner with respect to local neighborhoods. I included a specific mention about LLE connected to the analysis of MAP IT concerning local neighborhoods in **Appendix C, Perspectives on MAP IT via local neighborhoods**. This is the new mention:
> > >
> > > *To some degree, this perspective resembles the local neighhborhood perspective in Locally Linear Embedding (LLE) (Roweis and Saul, 2000). Also here, local neighborhoods are reconstructed in the target space versus the input space via an eigenvector operation. LLE and many other spectral methods (as mentioned in the Introduction of this paper) have inspired much of the research in visualization and neighbor embedding methods but as stated in Wang et al. (2021) in their section **Local structure preservation methods before t-SNE** (page 6) "the field mainly moved away" from these types of methods. In the future, it would be interesting to investigate closer links and differences between the fundamental LLE idea and MAP IT.*
> > >
> > > As explicitly stated in the title **"MAP IT to visualize representations"** the focus in this paper is on visualization. This is similar to the original t-SNE paper by van Maaten. Visualizations are provided. Quantitative analysis is of course important, but I feel that in a paper like this emphasizing visualization, that it is not a main point per se (but especially important to investigate more for downstream tasks). Especially in a paper such as UMAP, which was developed as a method to be used for downstream tasks in addition to visualization and for higher dimensional target spaces, quantitative analysis becomes more important.
> > >
> > > Please also note that the quantitative analysis experiment's conclusion was that by this analysis, MAP IT may seem better at capturing cluster structure at the potential expense of capturing manifold structures, and not the other way around as it may seem as you suggest.

---

### Meta-Review · Area_Chair_gu7R · 2023-12-10

**Metareview:**

The paper proposes a new way of visualization by using projective divergence as a cost function.

Strengths:
1) Elegant and clear formulation of the distance between distributions, and the cost function, that has projective property
2) Paper is wel-written

Weaknesses:
1) Computational complexity: $O(N^2)$  does not seem very scalable.
2) The formulation itself: the algorithms forms pairwise distances as in t-SNE, and then sums them up getting $p_i$, which should be very uniform; this questions the mathematical validity of the derivation

**Justification For Why Not Higher Score:**

It is much more of a borderline paper.

**Justification For Why Not Lower Score:**

I read the paper myself and I like the style, idea and the derivation, so it can be accepted.

---

### Decision · Program_Chairs · 2024-01-16

Accept (poster)